# ON HARD EPISODES IN META-LEARNING

## ABSTRACT

Existing meta-learners primarily focus on improving the average task accuracy across multiple episodes. Different episodes, however, may vary in hardness and quality leading to a wide gap in the meta-learner's performance across episodes. Understanding this issue is particularly critical in industrial few-shot settings, where there is limited control over test episodes as they are typically uploaded by end-users. In this paper, we empirically analyse the behaviour of meta-learners on episodes of varying hardness across three standard benchmark datasets: CIFAR-FS, mini-ImageNet, and tiered-ImageNet. Surprisingly, we observe a wide gap in accuracy of around $50\%$ between the hardest and easiest episodes across all the standard benchmarks and meta-learners. We additionally investigate various properties of hard episodes and highlight their connection to catastrophic forgetting during meta-training. To address the issue of sub-par performance on hard episodes, we investigate and benchmark different meta-training strategies based on adversarial training and curriculum learning. We find that adversarial training strategies are much more powerful than curriculum learning in improving the prediction performance on hard episodes.

## 1 INTRODUCTION

Humans have a remarkable ability to learn new concepts from very few examples and generalize effectively to unseen tasks. However, standard deep learning approaches still lag behind human capabilities in learning from few examples. For large over-parameterized deep models, learning with general supervision from only a few examples leads to over-fitting and thus poor generalization. To circumvent this, the paradigm of few-shot learning (Wang et al., 2020; Fei-fei et al., 2006; Vinyals et al., 2017) aims to effectively learn new concepts from very few labeled examples. These learned concepts can generalize well to future unseen learning tasks. Several frameworks have been proposed for tackling the few-shot learning scenario: transfer-learning (Dhillon et al., 2019), self-training (Phoo & Hariharan, 2020) and meta-learning (Hospedales et al., 2020; Finn et al., 2017; Snell et al., 2017). Meta-learning in particular aims to learn the process of learning from few examples and has shown remarkable performance across various few-shot benchmarks (Hospedales et al., 2020). In meta-learning, several few-shot tasks (episodes) are sampled from a set of base classes and the underlying model is trained to perform well on these tasks leading to improved generalization in learning from only few examples belonging to novel and unseen classes.

Existing meta-learners such as prototypical networks (Snell et al., 2017), MAML (Finn et al., 2017), MetaOptNet (Lee et al., 2019), and R2D2 (Bertinetto et al., 2018) primarily focus on improving prediction performance *on average* across multiple episodes. However, different episodes have distinct characteristics and hardness which might lead to a wide variance in prediction accuracy across episodes. This problem is much more prevalent in few-shot models deployed in the industry. For example, meta-trained models are often deployed in the cloud for the end-users to use for various tasks such as object recognition, detection, semantic segmentation in computer vision and natural language understanding in NLP. In such settings, the end-users upload their own few-shot dataset to perform predictions on new and unseen examples belonging to novel classes. In practice, different users may upload few-shot datasets of varying quality and hardness, leading to a wide disparity in performance across different users. To draw a parallel to the widely accepted experimental protocols in meta-learning, each of the uploaded few-shot dataset and the corresponding unseen examples is equivalent to a test episode.

In this paper, we study this issue and investigate how existing state-of-the-art meta-learners (Snell et al., 2017; Bertinetto et al., 2018; Lee et al., 2019) perform on episodes of varying hardness. Across three benchmark datasets: CIFAR-FS, mini-ImageNet, and tieredImageNet, we observe that there is a gap of $\approx 50\%$ in prediction accuracy between the easiest and hardest episodes. To this end, we identify several intriguing properties of hard episodes in meta-learning. For instance, we find that hard episodes are *forgotten* more easily than easy episodes during meta-training. Episode *forgetting* occurs when the underlying meta-learner forgets acquired knowledge by the end of the meta-training. To improve prediction performance on hard episodes, we investigate and benchmark various adversarial training and curriculum learning strategies that can be used jointly with any existing meta-learner. Empirically, we find that adversarial training strategies are much more powerful than curriculum learning in improving the prediction performance on hard episodes. The aim of our paper is not to chase another state-of-the-art in meta-learning, but to perform a fine-grained inspection of hard episodes across various meta-learning methods.

In summary, we make the following contributions:

- We present a detailed analysis of episode hardness in meta-learning across few-shot benchmarks and state-of-the-art meta-learners. In particular, we study various properties (e.g., semantic characteristics, forgetting) of episode hardness across different meta-learners and architectures.

- We find strong connections between episode hardness and catastrophic forgetting in meta-learning. While catastrophic forgetting can occur when meta-training with multiple datasets in sequence (Yap et al., 2020), we observe that forgetting events can occur even when the tasks during meta-training are drawn from a single dataset. In particular, we find that hard episodes are easy to forget, while easy episodes are difficult to forget.

- Based on our analysis, we investigate and benchmark different adversarial training and curriculum training strategies to augment general purpose meta-training for improving prediction performance on hard episodes. Empirically, we find that although there is no one-size-fits-all solution, adversarial meta-training strategies are more powerful when compared to curriculum learning strategies.

## 2 BACKGROUND AND RELATED WORK

Meta-learning aims to learn an underlying model that can generalize and adapt well to examples from unseen classes by the process of learning to learn. This is primarily achieved by mimicking the evaluation and adaptation procedure during meta-training. In general, there are three types of meta-learners: (a) Memory-based methods (Ravi & Larochelle, 2017; Munkhdalai et al., 2018; Santoro et al., 2016) adapt to novel classes with a memory attached to the meta-learner; (b) Metric-learning based methods (Snell et al., 2017; Sung et al., 2017) aim to learn transferable deep representations which can adapt to unseen classes without any additional fine-tuning; (c) Optimization based methods (Finn et al., 2017; Lee et al., 2019; Bertinetto et al., 2018) learn a good pre-training initialization for effective transfer to unseen tasks with only a few optimization steps. Although the primary focus of our work is meta-learning, we note that other few-shot learning paradigms such as transfer learning (Chen et al., 2021; Sun et al., 2019; Dhillon et al., 2020) have also shown competitive performance with meta-learning.

While there has been a significant progress in improving the state-of-the-art in meta-learning, very few work investigates the effectiveness of existing meta-learning approaches on episodes of varying hardness. A recent and concurrent work by Arnold et al. (2021) discusses episode difficulty and the impact of random episodic sampling during meta-training. Based on their analysis, Arnold et al. (2021) propose a re-weighted optimization framework for meta-training based on importance sampling. Although our paper and Arnold et al. (2021) tackle similar problems of episodic hardness, there are several points which distinguishes our work:

- We provide a much more fine-grained analysis of episode hardness than Arnold et al. (2021). Arnold et al. (2021) primarily discuss the transferability of episodes across different meta-learners, while we find and investigate a strong connection between episode hardness and catastrophic forgetting.

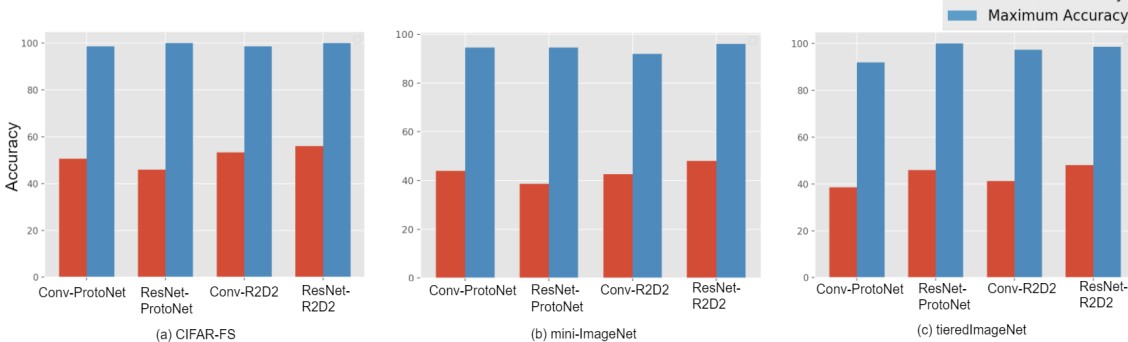

Figure 1: Accuracy (y-axis) of existing meta-learners on the hardest and the easiest episode across standard few-shot datasets and meta-learners (x-axis). Note that there is a wide gap of $\approx 50\%$ between the prediction performance on the easiest and hardest episode.

- Arnold et al. (2021) propose a loss re-weighting framework for improving the average accuracy across episodes. In contrary, we investigate the effectiveness of adversarial training (Gong et al., 2020) and general curriculum learning techniques in improving the average as well as worst-case prediction performance in meta-learning.

Adversarial meta-learning techniques have previously been used in conjunction with data-augmentation (Ni et al., 2021) to select the augmentation type resulting in the worst-case loss among different augmentation techniques. In this paper, we focus on how such strategies can be useful in improving the prediction performance of the hard episodes in addition to the average accuracy.

## 3 RETHINKING EPISODIC ACCURACY

Existing state-of-the-art meta-learners (Finn et al., 2017; Lee et al., 2019; Snell et al., 2017; Bertinetto et al., 2018) primarily focus on optimizing for the average loss across multiple training episodes or tasks. However, solely the average performance in isolation does not give enough insights into how meta-learners perform on episodes of varying quality and hardness. Such insights can be particularly crucial to investigate and debug meta-learning models deployed in the wild, where the model can encounter diverse test episodes. In this section, we go beyond the average accuracy across different test episodes and evaluate meta-learners on episodes of varying hardness. First, we discuss how to quantify the hardness of an episode and then discuss the performance of meta-learners on hard episodes.

### 3.1 WHAT IS A GOOD MEASURE OF EPISODE HARDNESS?

Episodic sampling (i.e. sampling various few-shot tasks from a base dataset) in meta-learning takes place in two steps: (i) First the episode classes are sampled from the class distribution of the base classes : $c \sim p(\mathcal{C}_{base})$; (ii) Next, an episode $\tau$ is sampled from the data distribution conditioned on the set of sampled classes $c$: $\tau \sim p(\mathcal{D}|c)$, where $\mathcal{D}$ is the base dataset. An episode $\tau$ consists of a set of support examples $\tau_s$ and query examples $\tau_q$. In few-shot learning, a $n$-way, $k$-shot episode is sampled which results in sampling $n$ classes and $k$ support examples per class. Based on this, the meta-learning optimization objective can be generalized as the following:

$$\theta^* = \arg\min_\theta \mathbb{E}_\tau[\ell(\mathcal{F}_{\theta'}, \tau_q)] \qquad (1)$$

where $\mathcal{F}$ is the base architecture with $\theta$ as the model parameters and $\theta' = \mathcal{A}(\theta, \tau_s)$ is the fine-tuning step with the support examples. Different meta-learners have different types of fine-tuning procedures and we direct the readers to (Finn et al., 2017; Snell et al., 2017; Bertinetto et al., 2018) for more information on the characteristics of $\mathcal{A}$. Based on this definition, we define the hardness of an episode $\mathcal{H}(\tau)$ in terms of the loss incurred on the query examples in an episode:

$$\mathcal{H}(\tau) = \ell(\mathcal{F}_{\theta^*}, \tau_q) \qquad (2)$$

We choose query loss as a metric for hardness because of its inherent simplicity in computation as well as interpretation. In addition, we find a strong negative correlation between the episodic loss and the accuracy ($\approx -0.92$ for mini-ImageNet and $\approx -0.89$ for tieredImageNet with prototypical networks). This is true for other meta-learners such as R2D2 too (See Appendix A for more details). Alternatively, hardness of an episode can also be defined as the average log-odds of the query example (Dhillon et al., 2020).

## 3.2 PERFORMANCE OF META-LEARNERS ON HARD EPISODES

To understand the effectiveness of meta-learners on episodes of varying hardness, we first order the test episodes in decreasing order of their hardness. Then, we evaluate different meta-learners on the easiest and the hardest test episode. Across all the few-shot benchmark datasets such as mini-ImageNet, CIFAR-FS and tieredImageNet, we find in Fig. (1) that there is a gap in accuracy of $\approx 50\%$ between the episodes with the highest and the lowest loss. Furthermore, we find that the meta-learner and architecture which performs well on average, does not necessarily perform well on the hardest episode. For example, in the case of mini-ImageNet, prototypical networks with a stronger architecture such as ResNet performs better than the 4-layered convolutional architecture on an average, but not on the hard episodes. Moreover in Fig. (1), we notice that for tieredImageNet, prototypical networks with ResNet performs the best on easy episodes, while R2D2 with ResNet performs slightly better on the hard episodes. The wide gap in accuracy between hard and easy episodes can be magnified for meta-learners deployed in the wild, where the model might encounter episodes which are significantly hard and diverse in nature. Going forward, we believe that meta-learning studies should not only report average episodic accuracy, but also the prediction accuracy on easy and hard episodes, to present a complete picture of the effectiveness of the meta-learner.

## 4 VISUAL SEMANTICS OF HARD EPISODES

Based on the observed disparity in performance between hard and easy episodes, a natural question arises: what causes certain episodes to incur high loss? While quantitatively, the hardness of an episode can be defined in terms of the loss incurred on the query examples, it does not provide salient insights into the qualitative nature of hard episodes. In general, we find that episodes incur high loss when there is a mismatch in the semantic characteristics between the support and query examples. For example, when a majority of the support examples have objects of only one category in the frame and the query examples have multiple objects of different categories surrounding the primary object of interest, the underlying meta-learner often leads to a wrong prediction. Furthermore, when the shape of the objects in the query examples is slightly different from the objects in the support examples, the prediction is often erroneous.

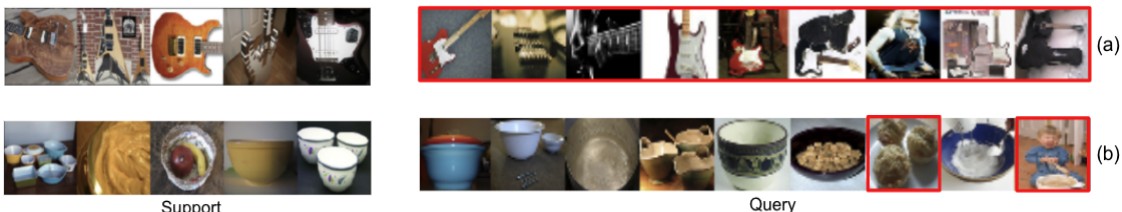

Figure 2: Semantic Properties of Hard and Easy Episodes: (a) Hard episode, Class: electric-guitar; (b) Easy episode, Class: mixing-bowl; The images marked in red borders are misclassified query examples.

In Fig. (2)-(a), we notice that the query examples that have different objects (e.g., humans) along with the primary object (i.e., guitar) are often misclassified. In Fig. (2)-(b), where most of the query examples are classified correctly, we find that the misclassified examples are of two types: (i) query images in which the primary object is occluded with a secondary object; (ii) the shape of the object in the query example is different from the object shapes in the support examples. We provide additional examples of hard episodes in Appendix E.

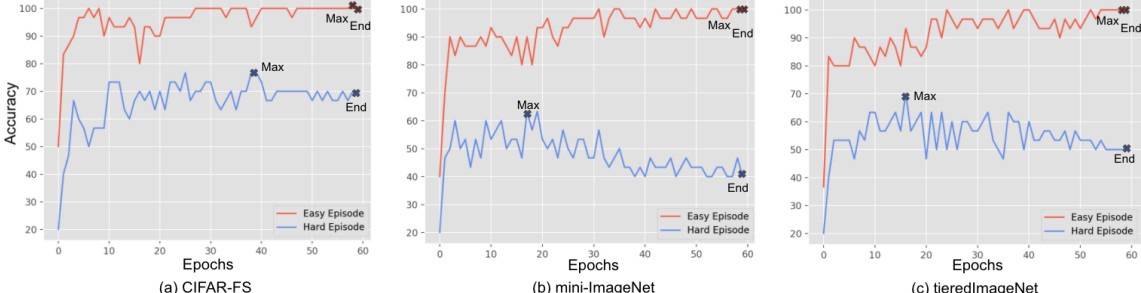

Figure 3: Accuracy of easy and hard episodes (y-axis) during the course of meta-training across different epochs (x-axis); Hard episodes often have a final accuracy less than the maximum accuracy reached during meta-training. The meta-learner used is prototypical networks with a Conv-4 backbone trained using episodes with 5-shot, 5-way.

# 5 HARD EPISODES SUFFER FROM FORGETTING

In supervised learning, catastrophic forgetting is prevalent when tasks from different distributions are learned sequentially (Kirkpatrick et al., 2016). In such cases, old tasks are forgotten as the model encounters and learns from new tasks. Toneva et al. (2019) has shown that certain examples can be forgotten with high frequency during the course of supervised training even when the samples are drawn from a single dataset. In meta-learning, Yap et al. (2020) has shown that in meta-training with tasks from *multiple* task distributions sequentially, tasks from the old distribution in the sequence can be forgotten as the meta-learner encounters new tasks. However, we observe that even in the case of meta-training with tasks drawn from a *single* task distribution, certain types of tasks (episodes) can be forgotten during the course of training. In particular, we analyze the connection between the hardness of episodes and catastrophic forgetting in meta-learning. We track the behaviour of easy and hard *training* episodes during meta-training and in summary find that:

(i) For hard episodes, we notice that the final accuracy at the end of the training drops significantly ($\approx 15\%$ in some cases) from the maximum accuracy obtained during the course of meta-training.

(ii) Hard episodes have more number of forgetting events in comparison to the easy episodes during the course of meta-training. This behaviour is more pronounced in the later stages of meta-training where the accuracies of the easy episodes have already stabilized.

## 5.1 DEFINING FORGETTING EVENTS IN META LEARNING

During the course of meta-training, the set of sampled *training* tasks are different in each epoch. In order to track forgetting events during meta-training, we first randomly select a set of $k$ episodes ($\mathcal{E} = \{\tau\}_{i=1}^{k}$) and track their accuracy, throughout the course of meta-training. In our experiments, we set $k = 160$ . We primarily define two types of forgetting events: (i) Local forgetting event; (ii) Global forgetting event.

**Global forgetting events**. For a given episode, a global forgetting event is encountered if the accuracy of the episode at the end of meta-training is less than the maximum accuracy reached during the course of training by a particular threshold. Formally, given an episode $\tau$ with the maximum accuracy $acc_{max}(\tau) = \max_j acc_j(\tau)$, a global forgetting event occurs if $acc_{max}(\tau) \geq acc_{end}(\tau) + \alpha$, where $\alpha$ is a threshold and $acc_{end}(\tau)$ is the accuracy at the end of meta-training. Note that for each episode, a global forgetting event can occur only once.

**Local forgetting events**. For an episode $\tau$ in the $j^{th}$ epoch of meta-training, a local forgetting event is encountered if the accuracy of the episode at the $j^{th}$ epoch ($acc_j(\tau)$) is less than the accuracy at the $(j-1)^{th}$ epoch ($acc_{j-1}(\tau)$) by a particular threshold, denoted by $\alpha$. Formally, a local forgotten event is encountered if $acc_j(\tau) + \alpha \leq acc_{j-1}(\tau)$. Empirically, we study local forgetting events for $0.03 \leq \alpha \leq 0.15$.

## 5.2 FORGETTING EVENTS AND HARD EPISODES

**Global forgetting events.** In Fig. (3), we track the accuracy of the hardest and easiest *training episode* from each of the few-shot datasets during the entire course of meta-training across different epochs. Visually, we observe that for the hard episode, the accuracy decreases after a certain point during the course of meta-training. However for the easy episode, this is not the case and the accuracy increases till the end of meta-training. To draw more insights, we compute the global forgetting behaviour of different episodes. We first choose the 15 hardest and easiest episodes respectively from CIFAR-FS, mini-ImageNet, tieredImageNet and compute their final episodic accuracy and the maximum episodic accuracy reached during meta-training. In Fig. (4), we find that for the hard episodes, the gap between the final accuracy and the maximum accuracy reached during meta-training is significantly larger than the easy episodes. The gap in particular is large for mini-ImageNet and tieredImageNet, while for CIFAR-FS the gap is relatively narrow. For example, in case of the mini-ImageNet dataset, the gap can be $\approx 15\%$, whereas for the tieredImageNet dataset, this gap can be $\approx 10\%$. For CIFAR-FS, this gap is $\approx 6\%$. This characteristic shows that hard episodes are globally forgotten in comparison to the easier episodes during the entire course of meta-training. We provide further results on global forgetting events in Appendix C.

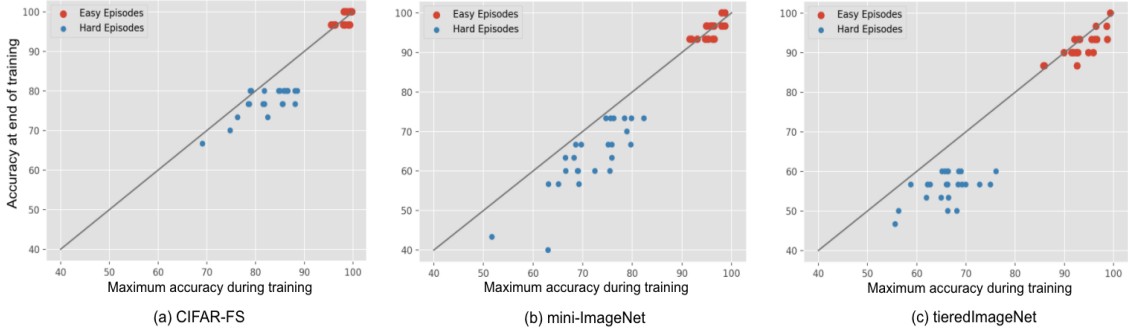

Figure 4: Hard episodes have a wider gap between the final accuracy (y-axis) and the maximum accuracy reached during meta-training (x-axis), in comparison to easy episodes. This behaviour is more pronounced for mini-ImageNet and tieredImageNet.

**Local forgetting events.** In order to compute the frequency of local forgetting events for easy and hard episodes across the three few-shot datasets, we first choose 15 easy and hard episodes from each dataset. Across this set of easy and hard episodes, we then compute the number of local forgetting events across various thresholds. In general, across the entire duration of meta-training, we find that the hard episodes have more local forgetting events than the easy episodes. For instance, in Fig. (5)-(a), we observe that there is a substantial gap in the number of forgetting events encountered for easy and hard episodes during the entire course of meta-training. Furthermore, to gain more insights about this gap in the number of encountered forgetting events, we understand how this gap behaves in the first 20 epochs of meta-training (Fig. (5)-(b)) and the last 20 epochs of meta-training (Fig. (5)-(c)). In particular, we find that the gap is narrow during the initial stages of meta-training, whereas the gap widens substantially during the later stages.

To summarize, we find that forgetting occurs in meta-learning even when the tasks are drawn from a *single* task distribution. Furthermore, we find a strong connection between episode hardness and forgetting events, where we show that hard episodes are more easily forgotten than easy episodes, both in the local and global contexts. In the next section, we investigate two meta-training strategies to improve few-shot performance on hard episodes.

## 6 IMPROVING PERFORMANCE ON HARD EPISODES

In this section, we investigate and benchmark two different meta-training strategies based on adversarial training and curriculum learning in order to improve prediction performance on hard episodes. Recent work (Gong et al., 2020; Ni et al., 2021) uses adversarial training to select episode specific

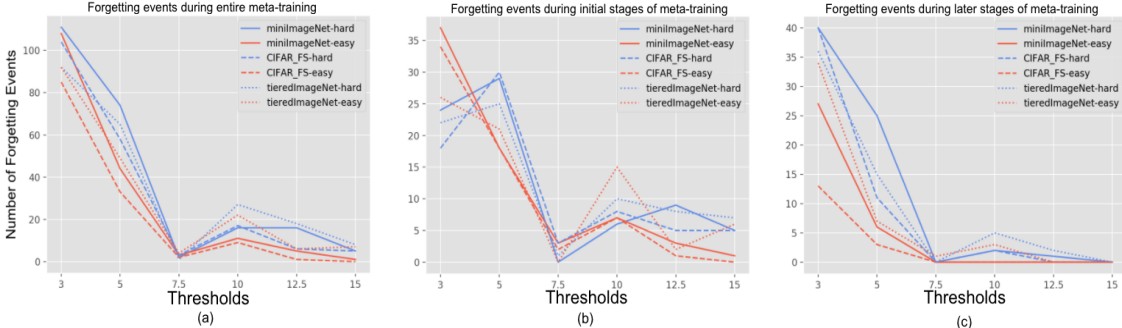

Figure 5: (a) Total number of local forgetting events (y-axis) across different thresholds (x-axis) during the course of meta-training; (b) Total number of local forgetting events during the first 20 epochs of meta-training; (c) Total number of local forgetting events during the last 20 epochs of meta-training. The number of local forgetting events is higher for hard episodes in comparison to the easy episodes across different thresholds.

augmentations from a wide pool of diverse data augmentation methods resulting in the highest loss. This loss is then optimized with respect to the model parameters during training. Such data augmentation selection strategies have been shown to mitigate over-fitting and improve generalization. In our work, we study how such training strategies can be used to select hard episodes and improve prediction performance across these episodes. In particular, we investigate two variants of adversarial training as proposed first in Gong et al. (2020): (i) General adversarial training (AT); (ii) Adversarial curriculum training (ACT).

## 6.1 GENERAL ADVERSARIAL TRAINING

We adopt the adversarial training procedure from (Gong et al., 2020) to first select episodes with a high loss and optimize the meta-learner only with respect to the loss incurred by such hard episodes. Specifically, this involves solving a saddle-point optimization problem where the underlying loss is minimized with respect to the parameters of the model and maximized with respect to the input. In particular, during each update of the meta-learner, we first draw a batch of episodes each containing support and query examples. Then for each element (episode) of the batch, we additionally sample new episodes with support and query examples belonging to the same classes as the original element of the batch. Then the episode with the highest loss is selected amongst the set consisting of the original episode of the batch and the additional sampled episodes. The model parameters are then updated with respect to the gradient of the loss incurred by the selected hard episode. Formally, we solve the following min-max optimization:

$$\min_{\theta} \mathbb{E}_{\tau}\big[\max_{t \in g(\tau)} \ell(\mathcal{F}_{\theta'}, t_q)\big] \tag{3}$$

where $\theta' = \mathcal{A}(\theta, t_s)$ is the fine-tuning step with the support examples from the selected episodes with the highest loss, $g(\tau)$ is an operator which samples additional episodes to select from for each task $\tau$ in the batch, $\mathcal{F}$ is the base learner and $\ell$ is the loss function. In our experiments, we let $g(\tau)$ select four additional episodes per sampled episode $\tau$. We provide a more detailed description of the hyper-parameters in Appendix B.1.

## 6.2 ADVERSARIAL CURRICULUM TRAINING

Curriculum learning (Hacohen & Weinshall, 2019; Bengio et al., 2009) aims to mimic the learning process of humans and animals. In particular, humans learn new tasks in a well defined order; i.e., it first learns easy tasks and then gradually moves towards learning more complex tasks. Inspired by this, we modify Eq. (3) and investigate a curriculum meta-training strategy. Specifically, during the initial phase of meta-training, we select only a set of easy episodes to learn from, while in the later stages of training, harder episodes are selected. Formally, given the underlying model is meta-trained for $|e|$ epochs, for the first $|e|/2$ epochs, the following loss is optimized:

$$\min_{\theta} \mathbb{E}_{\tau}\big[\min_{t \in g(\tau)} \ell(\mathcal{F}_{\theta'}, t_q)\big] \tag{4}$$

| Method | CIFAR-FS | | mini-ImageNet | | tieredImageNet | |
|---|---|---|---|---|---|---|
| | 1-shot | 5-shot | 1-shot | 5-shot | 1-shot | 5-shot |
| Conv-ProtoNet | 62.6 | 80.9 | 52.3 | 70.4 | 52.6 | 71.5 |
| + AT | **63.3** | 80.8 | **52.7** | **70.8** | 52.7 | **71.9** |
| + ACT | 63.1 | 80.0 | 52.4 | 70.3 | **52.9** | 71.5 |
| ResNet12-ProtoNet | 71.0 | 84.0 | 59.0 | 74.8 | 61.8 | 80.0 |
| + AT | **71.2** | 83.9 | 58.9 | **74.9** | 62.9 | **80.4** |
| + ACT | 69.3 | 82.6 | 56.0 | 72.8 | **63.4** | 79.9 |
| Conv-R2D2 | 67.6 | 82.6 | 55.3 | 72.4 | 56.8 | 75.00 |
| +AT | **68.0** | **82.8** | **55.7** | 72.4 | **57.2** | 75.00 |
| +ACT | **68.3** | 82.6 | 55.4 | 71.3 | **57.8** | 74.9 |
| ResNet12-R2D2 | 70.0 | 84.8 | 58.6 | 75.5 | 62.8 | 80.4 |
| +AT | **71.0** | 84.9 | 58.63 | **76.0** | **63.7** | **81.3** |
| +ACT | 69.4 | 83.2 | 56.3 | 74.4 | **63.5** | 80.9 |
| ResNet12 -MetaOptNet | 70.8 | 84.0 | 60.1 | 77.4 | 62.9 | 80.7 |
| +AT | **71.2** | **84.6** | **60.9** | **78.2** | **63.3** | 80.9 |
| +ACT | 70.2 | 83.9 | 59.8 | 77.1 | 62.1 | 79.8 |

Table 1: Average episodic performance of general adversarial training (AT) and adversarial curriculum training (ACT) across different meta-learners.

and for the last $|e|/2$ epochs during meta-training, the general min-max adversarial loss described in Eq. (3) is used.

# 7 EXPERIMENTS

## 7.1 EXPERIMENTAL SETUP

**Datasets**. We use three standard few-shot classification datasets for our experiments : (i) CIFAR-FS (Bertinetto et al., 2018); (ii) mini-ImageNet (Vinyals et al., 2016) and (iii) tieredImageNet (Ren et al., 2018). CIFAR-FS is sampled from CIFAR-100, whereas mini-ImageNet and tieredImageNet are subsets of ImageNet (Deng et al., 2009). We use the class splits from (Lee et al., 2019) for all the three datasets. Note that tieredImageNet is a more challenging dataset than mini-ImageNet as the splits are constructed from near the root of the ImageNet hierarchy. We provide more details on the datasets in Appendix B.2.

**Architectures**. We primarily use two standard architectures for our experiments: (i) 4-layer convolutional network as introduced in Vinyals et al. (2017); (ii) ResNet-12 (He et al., 2015) which is used by Oreshkin et al. (2018) in the few-shot setting. Both of these architectures use batch-normalization after every convolutional layer and use ReLU as the activation function. Similar architectures for few-shot learning have been previously used in Lee et al. (2019); Arnold et al. (2021).

**Meta-learners**. We use prototypical networks (Snell et al., 2017) from the metric learning family of few-shot algorithms. In addition, we use MetaOptNet (Lee et al., 2019) and R2D2 (Bertinetto et al., 2018) as representative algorithms from the optimization based meta-learning methods.

**Model Selection**. We use the validation set for each dataset to select the best model. Primarily, we run the validation procedure every 1k iterations on 2k episodes from the validation set to select the best model. Finally, we evaluate on 1k test episodes from the test set for each dataset.

## 7.2 DISCUSSION

### 7.2.1 ADVERSARIAL TRAINING IMPROVES PERFORMANCE ON HARD EPISODES

Across different meta-learners (ProtoNets, R2D2, MetaOptNet) and few-shot datasets (CIFAR-FS, mini-ImageNet, tieredImageNet), we find that the Adversarial Training (AT) strategy works well in general to improve both the average episodic performance as well as the episodic performance on hard episodes. The comprehensive results for the average performance of meta-learners is presented in Table (1), while the results on hard episodes are presented in Table (2). In particular, we find

| Method | CIFAR-FS | | mini-ImageNet | | tieredImageNet | |
|---|---|---|---|---|---|---|
| | 1-shot | 5-shot | 1-shot | 5-shot | 1-shot | 5-shot |
| Conv-ProtoNet | 28.8 | 63.0 | 20.2 | 53.2 | 22.2 | 52.3 |
| + AT | **29.3** | 62.9 | 19.3 | **53.5** | **24.6** | **52.7** |
| + ACT | 28.5 | 61.3 | 19.4 | 52.7 | 22.1 | **53.1** |
| ResNet12-ProtoNet | 36.0 | 66.7 | 26.2 | 56.5 | 29.8 | 60.4 |
| + AT | 33.7 | 66.7 | **29.2** | **58.3** | **31.2** | **60.9** |
| + ACT | 29.46 | 64.9 | 24.5 | 54.7 | 29.6 | 60.4 |
| Conv-R2D2 | 32.2 | 65.3 | 21.4 | 55.7 | 27.0 | 56.6 |
| +AT | **35.3** | **65.7** | **24.0** | 55.6 | **27.6** | 56.6 |
| +ACT | 32.8 | 65.5 | **23.7** | 54.5 | **27.8** | 56.7 |
| ResNet12-R2D2 | 30.6 | 68.0 | 28.0 | 59.0 | 34.0 | 61.5 |
| +AT | **35.6** | 68.0 | **28.5** | **60.1** | 33.8 | **62.1** |
| +ACT | 31.8 | 65.9 | 24.4 | 56.8 | 30.8 | **62.0** |
| ResNet12 -MetaOptNet | 37.2 | 69.2 | 29.5 | 61.1 | 30.4 | 61.5 |
| +AT | **37.9** | **70.0** | **31.2** | **62.3** | **31.5** | 61.4 |
| +ACT | 37.1 | 69.1 | 29.2 | 60.5 | 30.2 | 60.9 |

Table 2: Performance of general adversarial training (AT) and adversarial curriculum training (ACT) across different meta-learners on *hard episodes*. We report the mean accuracy over 30 hardest episodes for each meta-learner.

that adversarial meta-training strategies never hurt the average episodic performance and improves over the baseline in a majority of our experimental settings. However, we find that the Adversarial Training strategy (AT) leads to a large gain over the baseline meta-training strategy for hard episodes. Specifically, we find that the improvements are more significant for the 1-shot case when compared to the 5-shot case. For example, in the 1-shot case, we observe a $5\%$ gain for CIFAR-FS with R2-D2 and $3\%$ gain for mini-ImageNet with prototypical networks. For tieredImageNet, we observe $\approx 2\%$ improvement on episodic performance for hard episodes with prototypical networks.

### 7.2.2 ADVERSARIAL TRAINING IS BETTER THAN CURRICULUM TRAINING

Although curriculum training leads to better generalization in supervised learning (Hacohen & Weinshall, 2019; Bengio et al., 2009), we find that in meta-learning, the Adversarial Curriculum strategy (ACT) generally performs worse than both the baseline and general Adversarial Training (AT) in a majority of our experimental settings. Our observation on curriculum training for meta-learning is consistent with the recent work of (Arnold et al., 2021) where they show that curriculum meta-training strategies underperform significantly when compared to the baseline meta-training. We note that although the curriculum formulation in (Arnold et al., 2021) is different than ours, both methods present easy episodes to the meta-learner first followed by hard episodes. While we present a negative result on curriculum meta-training, we believe that this observation can be used as a guide to develop more advanced and improved curriculum meta-training strategies in the future. In summary, we find that although there is no one-size-fits-all solution to improve performance on hard episodes, adversarial meta-training strategies perform better than the baseline and curriculum learning.

## 8 CONCLUSION

In this paper, we investigated how different meta-learners perform on episodes of varying hardness. We found that there exists a wide gap in the performance of the meta-learners between the easiest and the hardest episode across different few-shot datasets. Furthermore, we investigated various facets of the hard episodes and uncovered two major properties: (i) Hard episodes usually have multiple diverse objects in the query examples, whereas the support set primarily consists of objects of a single category, (ii) Hard episodes are forgotten by the underlying meta-learner at a higher frequency than the easier episodes. To improve prediction performance over hard episodes, we investigated and benchmarked different training strategies such as adversarial training and curriculum learning during meta-training. We found that adversarial training strategies are beneficial towards meta-training and improve the average episodic performance as well as performance over hard episodes. Based on our

analysis in this paper, designing more robust meta-learning algorithms that can generalize to hard episodes is an important direction of future work.

## 9    REPRODUCIBILITY STATEMENT

Small experimental details are crucial to reproduce results in deep learning. In order to foster the reproducibility of our paper, we provide all the small details of our experiments in the Appendix (Section : Hyperparameters). We would like to point out that we use the general hyper-parameters in our experimental setup, as used in (Lee et al., 2019) to ensure reproducibility of the baselines which is particularly important in few-shot learning research. Additionally, considering a majority of our paper is on analysis of meta-learners and generating plots, we provide all the necessary details (e.g, number of episodes chosen for generating the plots) available in each corresponding section (See the subsections of Section 5).

## 10    ETHICS STATEMENT

In recent times, there has been a significant increase in the usage of deep learning models in the industry. Domains of application include NLP, computer vision and healthcare to name a few. Training task and domain specific deep models require large amounts of labeled training data which is not available for a variety of use-cases. The paradigm of few-shot learning (or meta-learning) aims to learn generalizable models from only a few training examples. Our paper is primarily geared towards understanding the science of such few-shot learning methods. We believe our work can accelerate research on few-shot learning with a specific focus on improving learning from hard episodes. One possible indirect risk is the deployment of our model without following the best practices for collecting fair and unbiased datasets for training. Our hope is that the positive impacts outweigh the negatives, specifically because standard best-use practices should mitigate most of the risks.

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

# A LOSS AS A MEASURE OF HARDNESS

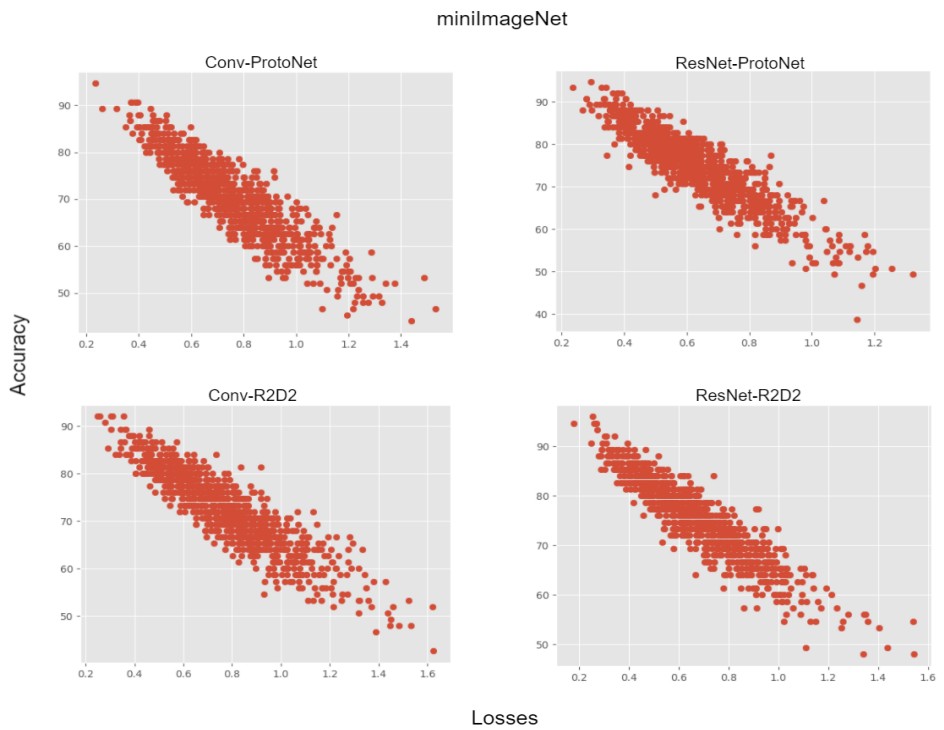

Figure 6: Loss (x-axis) vs. Accuracy (y-axis) plot for miniImageNet.

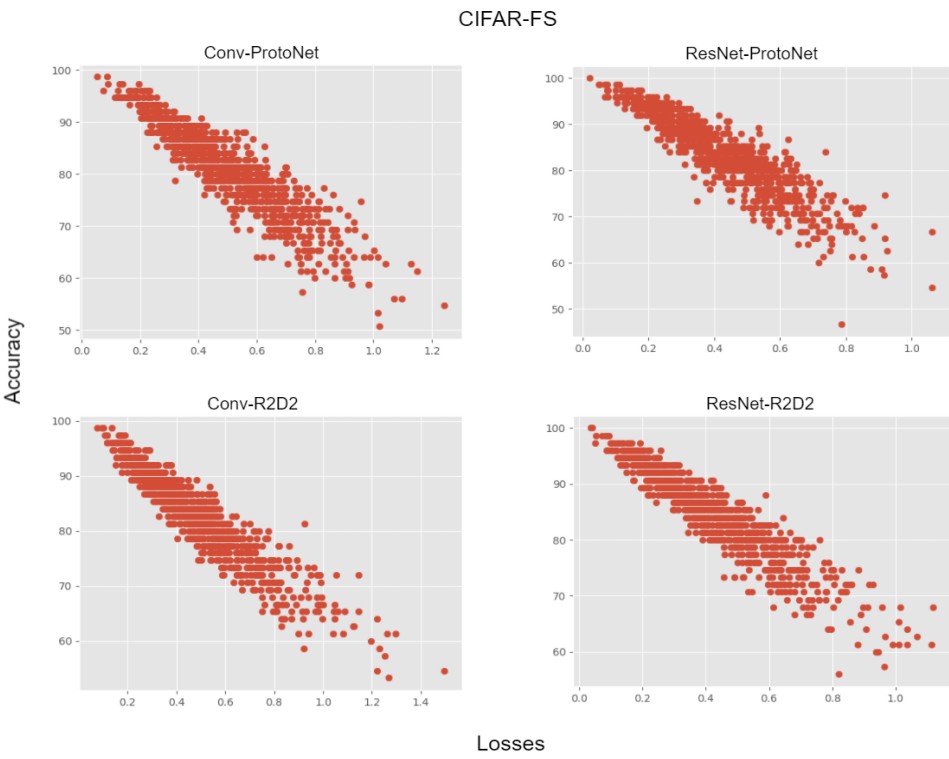

Figure 7: Loss (x-axis) vs. Accuracy (y-axis) plot for CIFAR-FS.

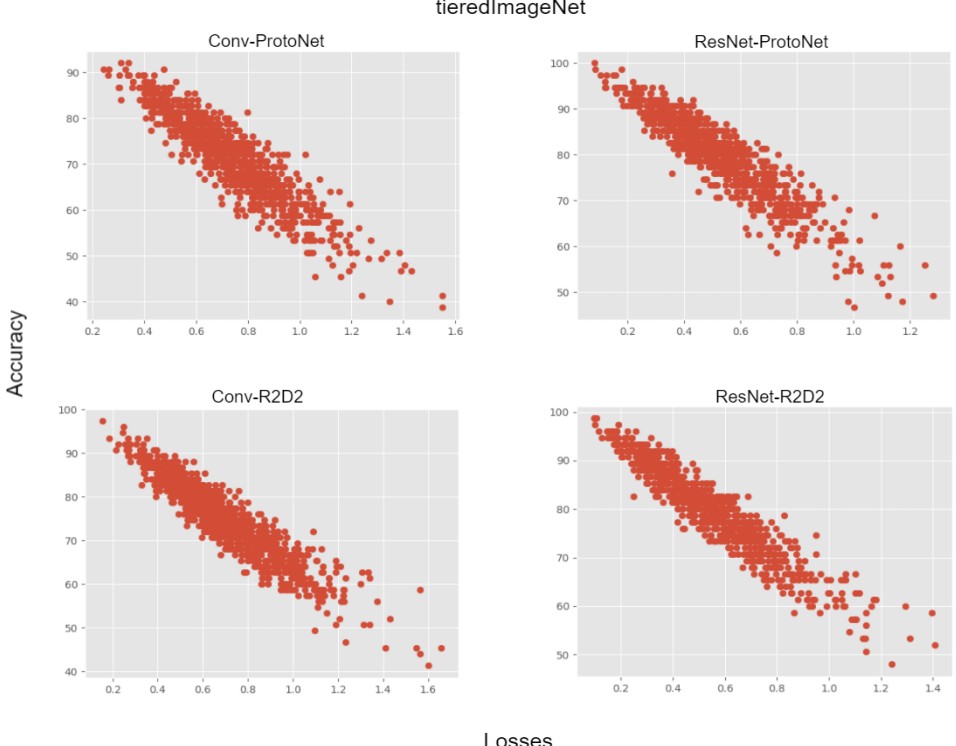

Figure 8: Loss (x-axis) vs. Accuracy (y-axis) plot for tieredImageNet.

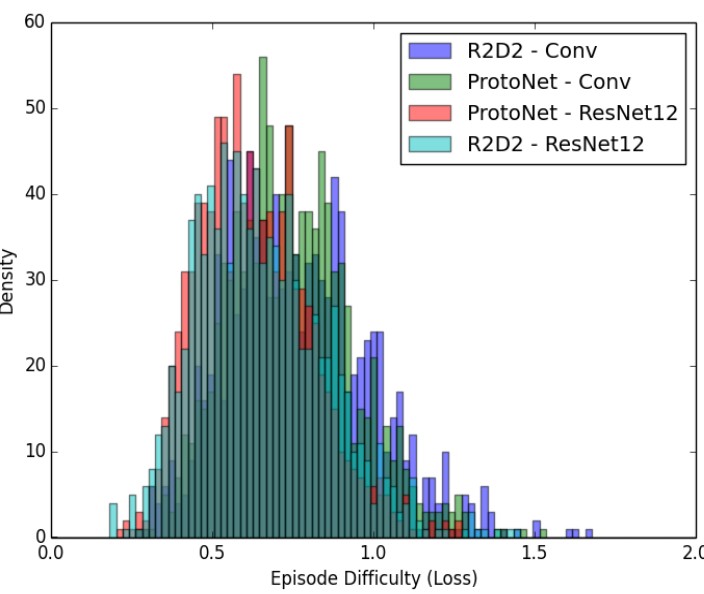

Figure 9: Distribution of episode hardness (difficulty) for mini-ImageNet across different meta-learners. We find that the distribution of hardness approximately follows a gamma distribution. From the density plots, we find that a significant fraction of the episodes lie in the range of medium-hard range of difficulty.

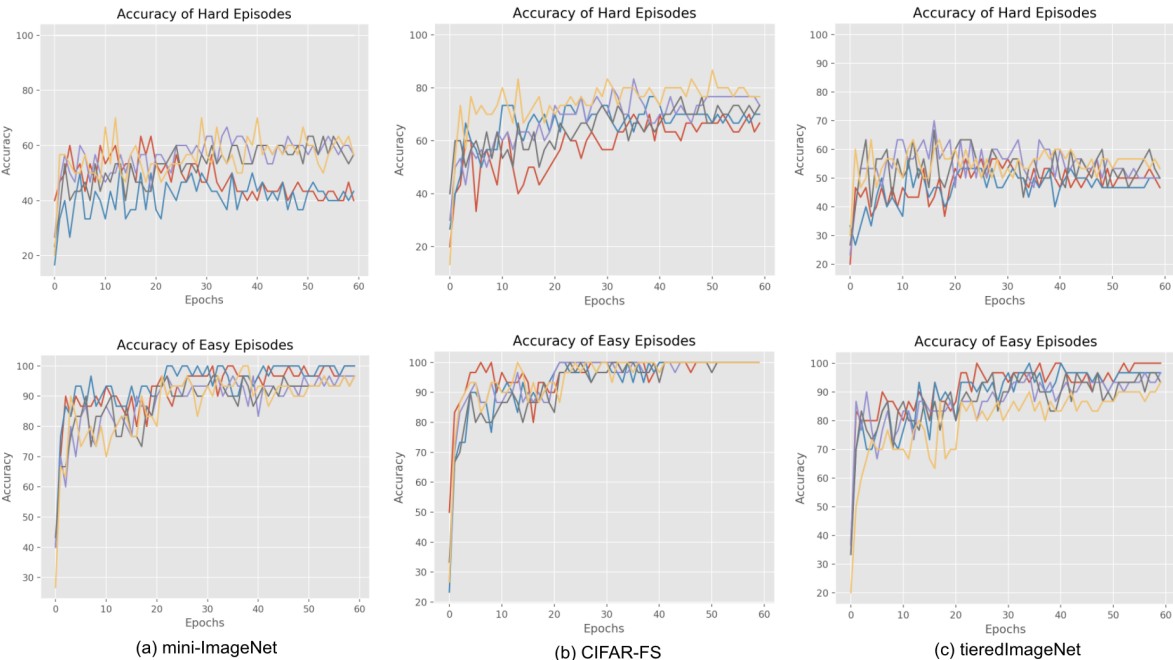

Figure 10: Accuracy of hard and easy episodes (y-axis) during the course of meta-training across different epochs (x-axis). Different colors signify different episodes.

## B IMPLEMENTATION DETAILS

### B.1 HYPERPARAMETERS

Similar to (Lee et al., 2019), for the optimizer we use SGD with Nesterov momentum and weight decay. We set the momentum to be 0.9 and weight decay to be 0.0005. During meta-training, the underlying meta-learner is trained for 60 epochs with each epoch consisting of 1000 episodes. We use 8 episodes per batch during the meta-learner update. The initial learning rate is kept at 0.1 which is then decayed using a learning rate scheduler at epoch 20, 40 and 50 as shown in (Lee et al., 2019). We use a 5-way classification for each of our experimental settings. During each iteration of meta-training, we use 6 query examples, while during meta-testing we use 15 query examples. Following the practice of (Gidaris & Komodakis, 2018; Qiao et al., 2017; Lee et al., 2019), we use horizontal flip, random crop, and color jitter as data augmentation techniques during meta-training. Across each of the experimental setup, the best model for meta-testing is chosen based on the best validation set performance. In general, few-shot learning can be performed in two scenarios: (i) Inductive setting where each test example is evaluated independently; (ii) Transductive setting where the few-shot learner has access to all the test examples. In all our experimental settings, we follow the inductive setting where each query example is evaluated independently. For general adversarial training method (AT), we choose sample 4 additional episodes per sampled episode. For a batch of size 8, there would be 32 additional episodes with a total of 40 episodes. 8 episodes with the highest loss are selected from this pool for optimizing the meta-learner. For our adversarial curriculum learning method (ACT), we choose the set of 8 episodes with the lowest loss for the first 30 epochs of meta-training and choose the set of 8 episodes during the last 30 epochs of meta-training. Across all our experimental settings, we use the same number of shots during meta-training and meta-testing to match the training and testing conditions.

### B.2 FEW-SHOT DATASETS

**mini-ImageNet .**    The mini-ImageNet dataset is introduced by (Vinyals et al., 2016) and is a standard few-shot classification benchmark. This dataset consists of 100 classes which are chosen from ILSVRC-2012 (Deng et al., 2009). The 100 classes are split into 64, 16, and 20 classes for

meta-training, meta-validation and meta-testing. Each class has 600 images resulting in a total of 60000 images for the entire dataset.

**tieredImageNet.** The tieredImageNet dataset is a larger and more challenging few-shot learning benchmark. It consists of 608 classes in total which are subclasses of ILSVRC-2012 (Deng et al., 2009). The total number of classes are split as 351, 97 and 160 classes for meta-training, meta-validation and meta-testing respectively. These classes are selected such that there is minimum semantic similarity between the different splits, which makes this dataset challenging.

**CIFAR-FS.** The CIFAR-FS dataset (Bertinetto et al., 2018) is curated from CIFAR-100 and comprises of 100 classes in total with each class consisting of 600 images. The classes are randomly split into 64, 16 and 20 for meta-training, meta-validation and meta-testing respectively.

Amongst the three datasets, the images are of size 84x84 for mini-ImageNet and tieredImageNet, while for CIFAR-FS, the images are of size 32x32.

## C  GLOBAL FORGETTING EVENTS

To gain further insights into global forgetting events, we select 5 hard and easy episodes from each of the few-shot datasets and track their accuracy during the course of meta-training. We use prototypical networks as the base meta-learner for our analysis. We find and validate in Fig. (10), that our analysis on global forgetting (as shown in the main paper) holds true for more number of hard episodes. Additionally, we also choose 25 hard and easy episodes to compute the mean of their maximum accuracy reached during meta-training and the accuracy at the end of training. We find that for hard episodes on an average, the gap between the maximum accuracy reached during training and the final accuracy is large. However, for easy episodes we find this gap to be narrow. This large gap for hard episodes signifies they they undergo catastrophic forgetting during the course of meta-training.

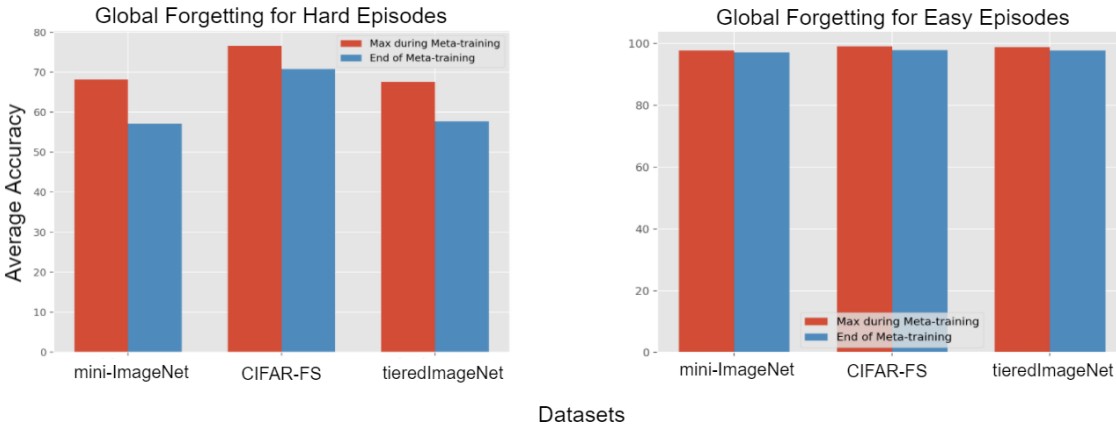

Figure 11: For a set of 25 hard and easy episodes, we report the mean of the maximum accuracy reached during the course of meta-training and the accuracy at the end of meta-training. For hard episodes, we observe a substantial gap denoting global forgetting.

## D  ON TRANSFERABILITY OF EPISODES

Recent and concurrent work such as (Arnold et al., 2021) shows that episode hardness transfer across different architectures and meta-learners. In this section, we revisit their analysis and take a closer look at the transferability of episodes across various meta-learners and architectures to obtain certain new insights. To this end, we compute the Pearson and Spearman correlations of the losses (hardness) incurred by episodes across different architectures and meta-learners for each few-shot dataset.

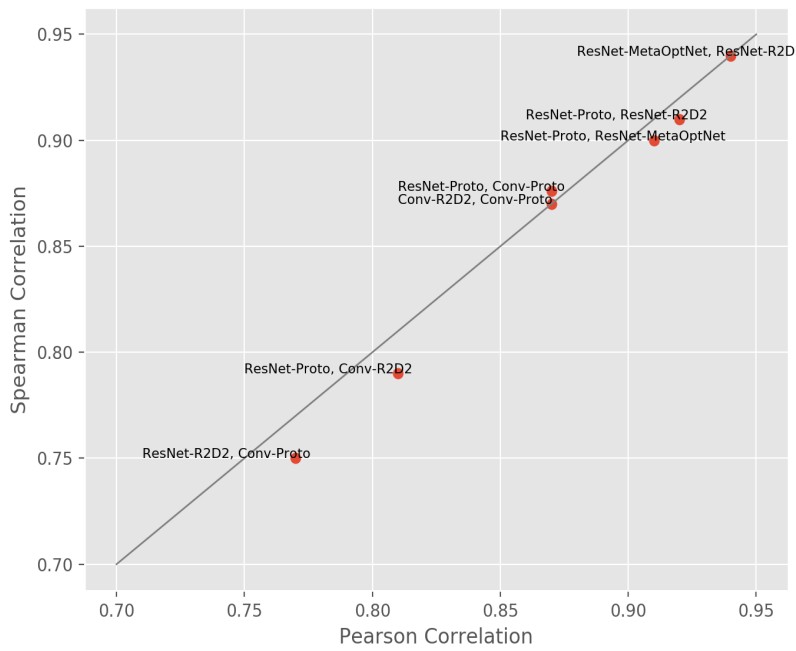

Figure 12: Transferability of episode hardness for tieredImageNet.

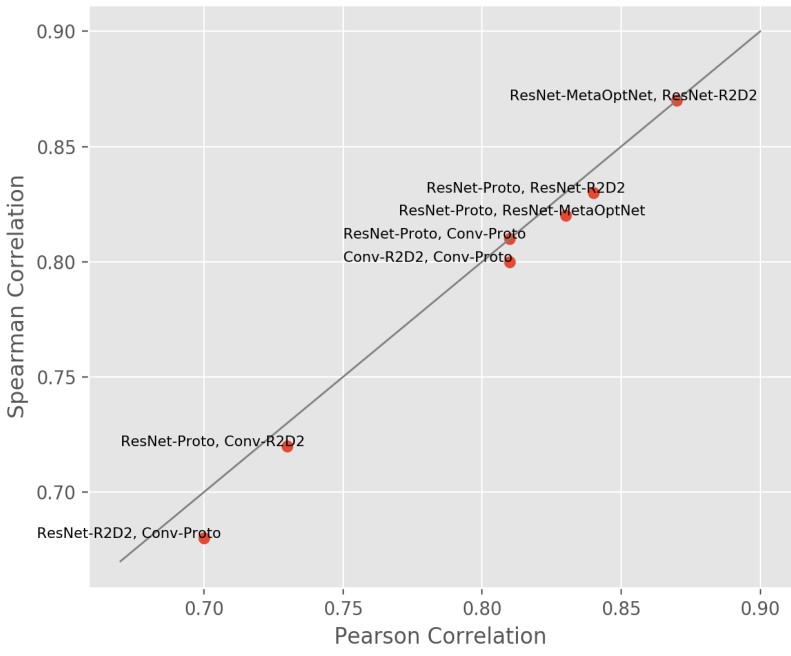

Figure 13: Transferability of episode hardness for mini-ImageNet.

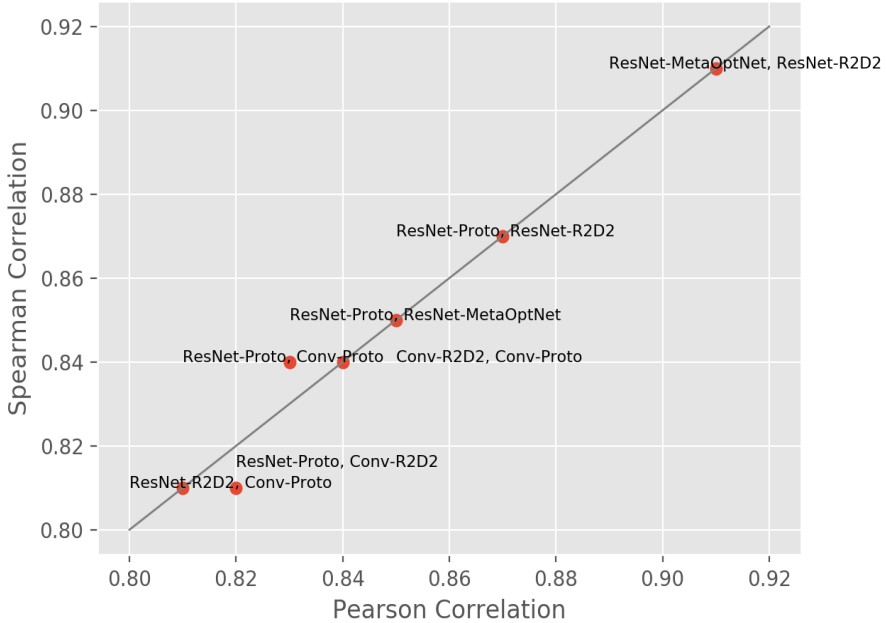

Figure 14: Transferability of episode hardness for CIFAR-FS.

Across all the three few-shot datasets, we find that the order of transferability across meta-learners and architectures is similar. For example, episodes transfer the best consistently between ResNet-MetaOptNet and ResNet-R2D2 across all the datasets, while episodes transfer the worst between ResNet-R2D2 and Conv-Proto. This observation makes intuitive sense as the meta-training strategies for both R2D2 and MetaOptNet are more similar for than R2D2 and prototypical networks. Amongst the datasets, we find that transfer of episode hardness between meta-learners and architectures is much more effective for tieredImageNet than mini-ImageNet. For example, out of the 7 combinations for transferability of episodes, 3 of them have a correlation of more than 0.90 in case of tieredImageNet.

## E  MORE ON VISUAL SEMANTICS OF HARD EPISODES

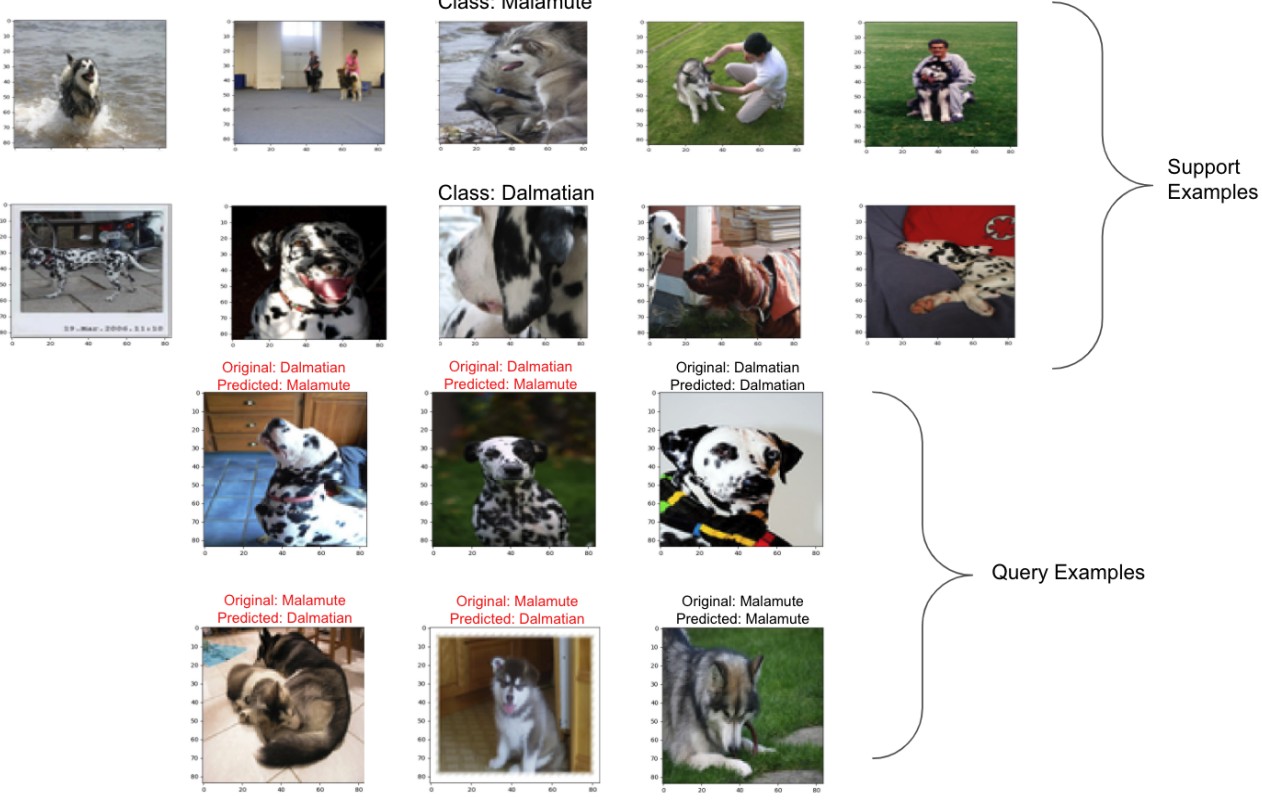

Figure 15: Visual semantic properties of hard episodes: We find that if there are closely related fine-grained categories in an episode (e.g, *Malamute* vs. *Dalmatian*), the meta-learner (prototypical networks + ResNet-12) often gives a wrong prediction.

In this section, we provide more instances of hard episodes. In particular, we find that episodes also become hard when there are closely related fine-grained categories in the episode. For example, in an episode consisting of dogs from the classes *Malamute* and *Dalmatian*, we notice that the underlying meta-learner often gets confused between the two fine-grained categories leading to erroneous predictions on the query examples. Presence of similar fine-grained categories in episodes is therefore one failure mode of existing meta-learning methods.

## F    DISTRIBUTION OF LOSSES FOR CIFAR-FS AND TIEREDIMAGENET

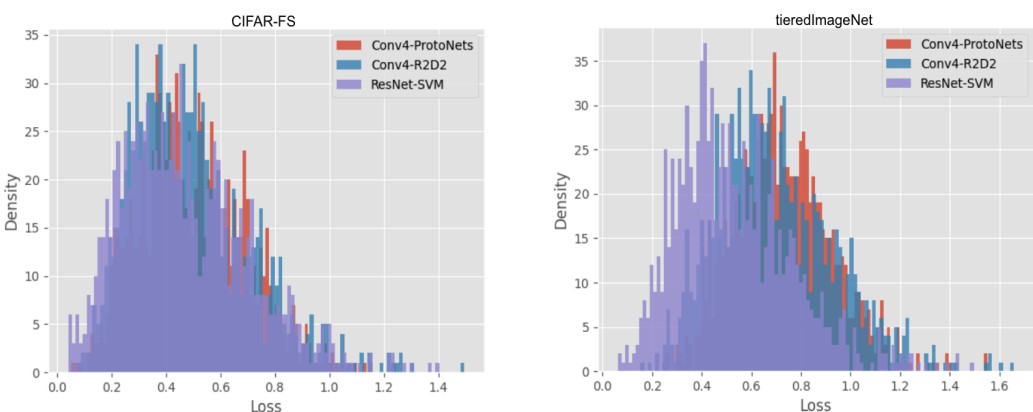

Figure 16:    Distribution of query losses for tieredImageNet and CIFAR-FS across Prototypical networks, R2D2 and MetaOptNet (SVM).

## G    FORGETTING EVENTS FOR R2D2 AND METAOPTNET

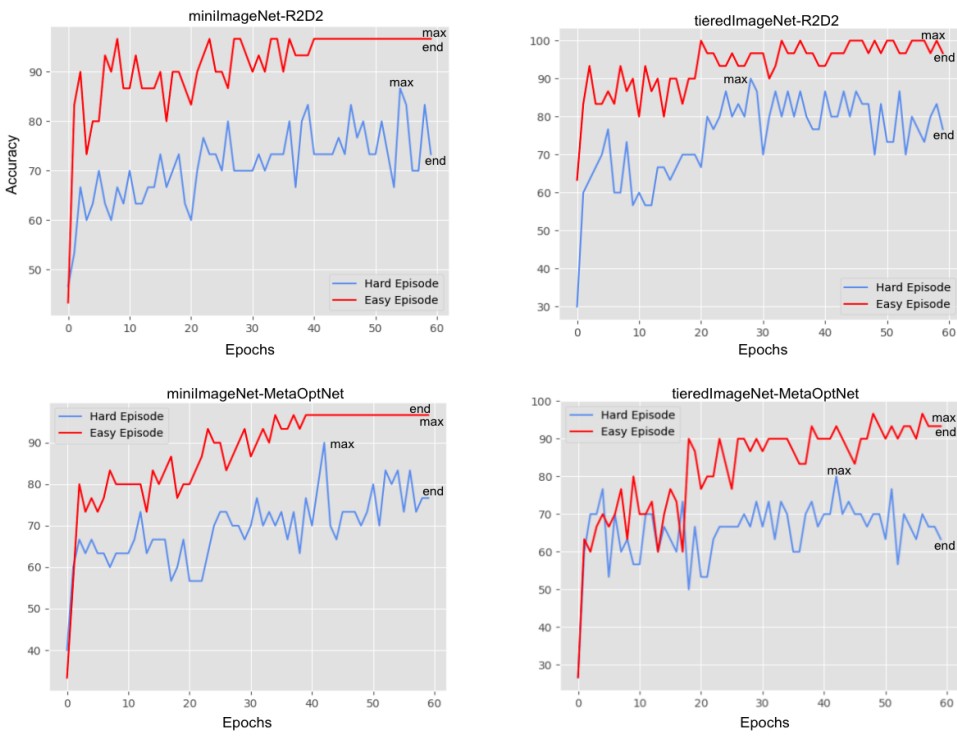

Figure 17:  Forgetting events for R2D2 and MetaOptNet across mini-ImageNet and tieredImageNet datasets.For R2D2 we use the Conv-4 backbone, while for MetaOptNet we use ResNet-12.

# H HARDNESS OF OOD EPISODES

Episode hardness and out-of-distribution (OOD) episodes have a strong connection between them. We emphasise that the primary aim of our paper is to analyse episode hardness when the episodes during meta-testing are drawn from the same task distribution as meta-training. However, we note that episodes can be hard for a meta-learner when the meta-test episodes are drawn from a different distribution. To analyse the hardness of OOD episodes, we first curate a new meta-testing set where the support examples in the new episodes are slightly perturbed. We use the standard corruptions used previously to generate the ImageNet-C dataset (Hendrycks & Dietterich, 2019) in order to create new episodes where the support examples are perturbed. Precisely, for every support example, we randomly choose from a set of perturbations: *(None, Gaussian Noise, Shot Noise, Impulse Noise, Blur, Frost, Fog, Brightness, Contrast, Elastic)* . This creates a new test episode which is OOD with respect to the episodes encountered during meta-training. From Fig. (18), we find that the query losses incurred for such perturbed or OOD episodes are higher than the in-distribution episodes. Notably, there is an overlap in the distribution of losses for the in-distribution and OOD episodes, which might indicate that some of the hard in-distribution episodes are actually outliers or OOD. An end-to-end study of the susceptibility of meta-learners to OOD episodes (and also training strategies to improve on OOD episodes) is an interesting research direction and an important direction for future work.

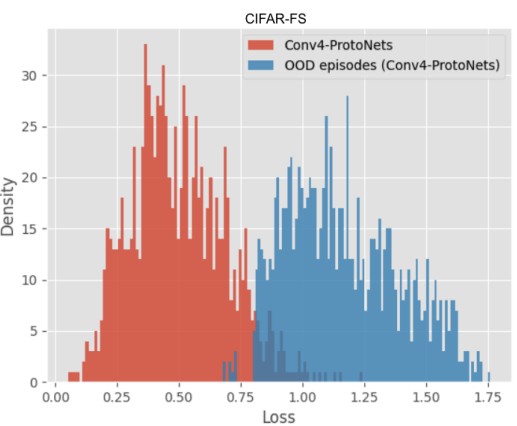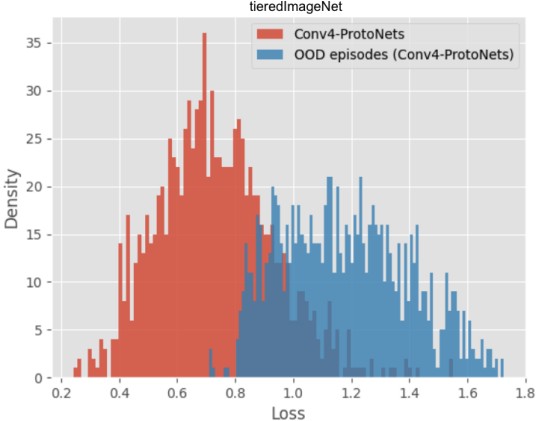

Figure 18: Distribution of query losses for in-distribution episodes (Red) and OOD episodes (Blue) for Prototypical Networks with a Conv-4 backbone.

