# OpenReview forum: "On Hard Episodes in Meta-Learning"
_ICLR.cc/2022/Conference — ICLR 2022 Submitted_

### Official Review · Reviewer_tHng · 2021-10-26

**Correctness:** 3
**Technical Novelty And Significance:** 3
**Empirical Novelty And Significance:** 3
**Recommendation:** 5
**Confidence:** 4

**Main Review:**

The investigated topic is interesting and relevant to the few-shot classification community, and the writing quality is good. I think going beyond the average-case in episodic evaluation has the potential to reveal useful insights, and I am happy to see work in that direction.

Some experimental details were not clear from the paper's description:

- **[major]** In Section 5, were easy and hard episodes sampled from training or test classes? This is a crucial distinction, and it determines whether I agree or not with the use of the term "forgetting". If episodes are sampled from test classes, then I have a hard time seeing how the model could forget what it has never seen throughout training. In that case, is "forgetting" simply another word for "meta-overfitting"?
- **[major]** What is the decision process for bolding entries in Tables 1–2? Was a statistical test for the significance of the difference in means performed? What are the 95% confidence intervals on the average query accuracies? Is a difference of 0.4% in query accuracy (e.g. Conv-ProtoNet+AT on mini-ImageNet) statistically significant? This is important, because quite a few entries in both tables show modest improvements over the baselines, and the claimed efficacy of adversarial training hinges on those improvements.
- I had to scroll down to Appendix B.1 to determine what "ways" value was used for Figures 1–5 (5-way), and I could not find what "shot" was used.
- Section 3.2 describes sorting the test episodes in decreasing hardness order, then evaluating various meta-learners on the easiest and hardest test episode. Which meta-learner was used to measure episode hardness? Were different easy and hard episodes chosen for different meta-learners?
- Which benchmark and meta-learner was used for Figure 2?
- Which meta-learner was used for Figures 3–5?
- How are "similar episodes" selected for adversarial training?

Additional questions/comments:

- **[major]** "For example, in the case of mini-ImageNet, prototypical networks with a stronger architecture such as ResNet performs better than the 4-layered convolutional architecture on an average, but not on the hard episodes." Can the authors clarify what numbers are used in drawing this conclusion? Is this based on Figure 1 and Table 1? Table 2 appears to contradict this statement.
- **[major]** In the strictest sense of the term, MetaOptNet and R2D2 can be characterized as optimization-based meta-learners, but Prototypical Networks could also be implemented with an optimization inner-loop rather than with the analytical solution for Gaussian classifiers, so is the distinction relevant? In all cases, the feature extractor remains frozen during adaptation. Would meta-learners that adapt the feature extractor (e.g. MAML, CNAPs) behave differently with respect to easy and hard episodes? In my opinion, this is an experimental blind spot for the submission.
- Going beyond the fixed-ways setting (like is done in Meta-Dataset, for instance), are there dynamic range issues arising when comparing query losses for episodes with various numbers of ways? Do the authors prescribe some procedure for normalizing the losses?
- The term "forgetting event" is used before being defined, which I found confusing.
- In Figure 5, what happens with threshold value 7.5? Why are the numbers of forgetting events collapsing towards zero for that specific threshold value?

**Summary Of The Paper:**

The submission investigates and characterizes episode difficulty (as defined in terms of its query loss) in few-shot classification and reports empirical observations on CIFAR-FS, mini-ImageNet, and tiered-ImageNet.

Across all benchmarks, a wide accuracy gap is observed between the hardest and easiest test episodes for multiple combinations of meta-learner (Prototypical Networks, R2D2) and network architecture (four-layer ConvNet, ResNet). The paper presents support and query images for easy and hard episodes and draws the conclusion that a mismatch in semantic or shape characteristics or in the number of objects between support and query images often cause misclassification.

The submission then examines how the accuracy of easy and hard episodes evolves over training. It uncovers global and local forgetting events and reports that the latter occur more frequently for hard episodes.

Finally, the paper examines two strategies for taking easy and hard episodes into account: adversarial training, and adversarial curriculum training. Adversarial training is shown to yield modest improvements over (and adversarial curriculum training does not perform significantly better than) regular training.

**Summary Of The Review:**

The submission examines a very interesting topic, but falls short on execution in its current state, especially in terms of experimental clarity and generalizability of the reported observations.

---

**Post-rebuttal**:  I appreciate the clarifications and think the paper has made good progress towards better clarity.

At this point, however, I'm not ready to reconsider my score, mainly because my concern with how results are reported remains unaddressed. In the absence of confidence intervals it's hard for me to determine whether the observed improvements are significant or not.

---

> ### Author Response · Authors · 2021-11-19
> **Response to reviewer tHng**
>
> **[major] In Section 5, were easy and hard episodes sampled from training or test classes? This is a crucial distinction, and it determines whether I agree or not with the use of the term "forgetting". If episodes are sampled from test classes, then I have a hard time seeing how the model could forget what it has never seen throughout training. In that case, is "forgetting" simply another word for "meta-overfitting"?**:   We want to clarify that the episodes used for analysis in Section (5) are sampled from the training classes used during meta-training. We have made these details more clear in the main paper (See updated Section (5)).
>
> **I had to scroll down to Appendix B.1 to determine what "ways" value was used for Figures 1–5 (5-way), and I could not find what "shot" was used**:  For Figures 1-5, we meta-trained using  5-shots per class. We have added this detail in both the main paper and Appendix.
>
> **Section 3.2 describes sorting the test episodes in decreasing hardness order, then evaluating various meta-learners on the easiest and hardest test episode. Which meta-learner was used to measure episode hardness? Were different easy and hard episodes chosen for different meta-learners?**:   In Section 3.2, the hardest and the easiest episodes were selected specific to the individual meta-learner. For e.g, for R2D2 + ResNet-12, we order the test-episodes based on the query loss incurred with R2D2 specifically to find the easiest and hardest episode.  We also note that (See Appendix (D)), that there is a satisfactory transferability of episode hardness across different meta-learners and architectures.
>
> **Which benchmark and meta-learner was used for Figure 2?** : For Fig. (2), prototypical network with a Conv-4 backbone is used for the miniImageNet dataset. We will add more visualizations for other meta-learners in the final version of our draft.  However note that episode hardness transfers satisfactorily across different meta-learners.
>
> **Which meta-learner was used for Figures 3–5?** : For Fig. 3-5, prototypical network with a Conv-4 backbone is used. We have added new experimental results on the connections between catastrophic forgetting and episode hardness for other meta-learners such as R2D2 and MetaOptNet in Appendix (G).
>
> **How are "similar episodes" selected for adversarial training?** : We clarify that by ‘similar’, we mean sampling different support and query examples belonging to the set of classes already sampled corresponding to each episode of the batch. We have also improved the writing in Section (6) and made the details more clear for the readers.
>
> **[major] For example, in the case of mini-ImageNet, prototypical networks with a stronger architecture such as ResNet performs better than the 4-layered convolutional architecture on an average, but not on the hard episodes. Can the authors clarify what numbers are used in drawing this conclusion? Is this based on Figure 1 and Table 1? Table 2 appears to contradict this statement** :  On an average, ResNet+Prototypical Network has an accuracy of 74.8 for the 5-shot, 5-way case, whereas Conv4 + Prototypical Network has an accuracy of 70.4.  However, if we look at the accuracy of each of the architectures only on the hardest episode, ResNet + Prototypical Network has an accuracy of 43% and Conv4 + Prototypical Network has an accuracy of 38% (See Fig. 1).  However, on a larger set of 30 hard episodes, a stronger backbone architecture such as ResNet gives better performance than a shallow backbone such as Conv4 (Table 2).
>
> **In the strictest sense of the term, MetaOptNet and R2D2 can be characterized as optimization-based meta-learners, but Prototypical Networks could also be implemented with an optimization inner-loop rather than with the analytical solution for Gaussian classifiers, so is the distinction relevant? In all cases, the feature extractor remains frozen during adaptation. Would meta-learners that adapt the feature extractor (e.g. MAML, CNAPs) behave differently with respect to easy and hard episodes**:   Thank you for the suggestion! We would first like to point out that although meta-learners such as MAML  fine-tune the feature extractor, it still underperforms other meta-learners such as R2D2 and MetaOptNet which keep the feature extractor fixed during meta-test. We have conducted two new experiments which fine-tunes the entire feature extractor: (a) Using MAML; (b) Using ERM (i.e, during meta-training, we train a feature backbone using supervised learning and during meta-testing, we fine-tune a linear head along with the feature extractor on the support examples). Across both the cases, we find that for few-shot methods, which adapt the feature extractor during meta-testing, there is a wide gap in the accuracy amongst the hardest and easiest episode.
>
> * ERM (5-shot, 5-way) : (Avg: 79.17, Min: 49.33, Max: 96.0)
> * MAML with Conv-4 (5-shot, 5-way) : (Avg: 60.13, Min: 29.3 , Max: 84.1)

---

> > ### Author Response · Authors · 2021-11-19
> > **More responses to reviewer tHng**
> >
> > **Going beyond the fixed-ways setting (like is done in Meta-Dataset, for instance), are there dynamic range issues arising when comparing query losses for episodes with various numbers of ways**:   In our experimental settings, the number of ways in each episode during meta-testing is constant (i.e 5). We have conducted two additional experiments addressing the reviewer’s question about dynamic number of ways:
> >
> > (a) Experiment 1: Meta-testing with number of ways different from the one used during meta-training, across all the episodes;
> >
> > * Format: {way: {avg accuracy, max accuracy, min accuracy}}
> > * {3: {88.7, 100, 43.4}; 5: {70.8, 92.0, 41.3}, 10: {55.97, 71.3, 36.6}; 15: {47.5, 59.11, 35.5}}
> >
> > (b) Experiment 2: Meta-testing with Meta-Dataset’s sampling procedure, where different sampled episodes have different values of way. In this case, we also find a wide gap in accuracy of ~45%, between the episodes with the highest and lowest accuracy, depending on the meta-learner.  Note that in this case, the query losses are already normalized (with respect to the total number of query examples in each episode).
> >
> > We also meta-train ResNet-12 with prototypical networks and R2D2 using the dynamic sampling procedure in Meta-Dataset and analyse their behaviour on both the easy and hard episodes.
> >
> > In general, we find that even training with a variable number of ways (using the method in Meta-Dataset) in each episode, there still exists a gap in the accuracy between the hard and easy episodes. In this sampling procedure, we dynamically sample both the classes and shots per class for support examples in each episode, keeping the query shots fixed. In summary, we find that even when meta-training with this dynamic sampling procedure, there is still a gap of ~40% accuracy between the easiest and hardest episode for both Prototypical Networks and R2D2.
> >
> > We will update the final version of our paper with these new experimental details.

---

> > > ### Comment · Reviewer_tHng · 2021-11-24
> > > **Rebuttal acknowledged**
> > >
> > > Thank you for your response. I appreciate the clarifications and think the paper has made good progress towards better clarity.
> > >
> > > At this point, however, I'm not ready to reconsider my score, mainly because my concern with how results are reported remains unaddressed. In the absence of confidence intervals it's hard for me to determine whether the observed improvements are significant or not.

---

### Official Review · Reviewer_uYEr · 2021-10-27

**Correctness:** 3
**Technical Novelty And Significance:** 2
**Empirical Novelty And Significance:** 2
**Recommendation:** 3
**Confidence:** 4

**Main Review:**


Strengths
---------

- Interesting analysis on the dynamics of hard vs easy tasks during training.
- Evaluation on adversarial and curriculum strategies is a plus.
- The authors recommend practitioners to report the prediction accuracy on easy and hard episodes. This is a sensible suggestion, since this metric is easy to report and can provide a glimpse on the robustness of the method.
- The paper is well written and easy to follow.

Weaknesses
----------

- No discussion about out-of-distribution tasks. It seems to me that there is a strong connection between hardness and out-of-distribution samples. Hard episodes could either be out-of-distribution samples or outliers that are on the margin of the decision boundaries. The authors do not dive into this link at all, neither in the related work section nor in the experiments. Note that, there has been substantial work on evaluating meta-learners on out-of-distribution episodes (Jeong et al. 2020, Lee et al. 2019, Wang et al. 2019). Unfortunately this is a strong weakness of the paper, since it completely disregard a line of work which is very relevant for the problem at hand. It is not easy to provide an actionable feedback on this point, since it will require substantial changes to the structure of the paper. I would like to see (i) a contextualization of hard tasks in the framework of out-of-distribution learning, (ii) experiments to evaluate the distance of hard tasks from the dataset distribution and (iii) evaluation of methods specifically designed to address out-of-distribution, e.g. OOD-MAML (Jeong et al. 2020).

- Limited number of conditions. For an experimental paper like this, I would expect to see an evaluation on more conditions. (i) In terms of methods, evaluation on methods developed for out-of-distribution learning (Jeong et al. 2020, Lee et al. 2019, Wang et al. 2019). Some Bayesian methods have also showed robustness against outliers and domain shift (Patacchiola et al. 2020). (ii) In terms of datasets, the Meta-Dataset (Triantafillou et al. 2019) has been recently proposed to evaluate meta-learner on more realistic conditions. The Meta-Dataset could be the perfect benchmark to explore the link between hardness and out-of-distribution samples. (iii) The authors compare adversarial training and curriculum learning strategies, but other strategies could also be tested. For instance, if the hardness of the episodes is known in advance, the loss could be balanced accordingly by using a scalar weight. This strategy would be similar in spirit to the Focal Loss (Lin et al. 2017).

- Nature of hard tasks. The authors qualitatively compare a few hard samples in Section 4 and Appendix E. However, it is difficult to draw some conclusions from this limited number of samples. It would be helpful to see a quantitative comparison based on clear metrics. For instance, in Section 4 the authors argue that wrong predictions are often due to multiple objects of different categories surrounding the primary object of interest. This hypothesis could be verified empirically by using few-shot datasets like ORBIT (Massiceti et al. 2021) that include a set of cluttered and clean images of the same objects. By managing the proportion of clean VS cluttered images in the task it should be possible to see if episodes get more or less hard.

References
----------

Arnold, S. M., Dhillon, G. S., Ravichandran, A., & Soatto, S. (2021). Uniform Sampling over Episode Difficulty. arXiv preprint arXiv:2108.01662.

Jeong, T., & Kim, H. (2020). OOD-MAML: Meta-learning for few-shot out-of-distribution detection and classification. Advances in Neural Information Processing Systems, 33.

Lee, H. B., Lee, H., Na, D., Kim, S., Park, M., Yang, E., & Hwang, S. J. (2019). Learning to balance: Bayesian meta-learning for imbalanced and out-of-distribution tasks. arXiv preprint arXiv:1905.12917.

Lin, T. Y., Goyal, P., Girshick, R., He, K., & Dollár, P. (2017). Focal loss for dense object detection. In Proceedings of the IEEE international conference on computer vision (pp. 2980-2988).

Massiceti, D., Zintgraf, L., Bronskill, J., Theodorou, L., Harris, M. T., Cutrell, E., ... & Stumpf, S. (2021). ORBIT: A Real-World Few-Shot Dataset for Teachable Object Recognition. arXiv preprint arXiv:2104.03841.

Patacchiola, M., Turner, J., Crowley, E. J., O'Boyle, M., & Storkey, A. (2020). Bayesian meta-learning for the few-shot setting via deep kernels.

Triantafillou, E., Zhu, T., Dumoulin, V., Lamblin, P., Evci, U., Xu, K., ... & Larochelle, H. (2019). Meta-dataset: A dataset of datasets for learning to learn from few examples. arXiv preprint arXiv:1903.03096.

Wang, K. C., Vicol, P., Triantafillou, E., Liu, C. C., & Zemel, R. (2019). Out-of-distribution Detection in Few-shot Classification.


**Summary Of The Paper:**

In this paper the authors present an investigation of the relations between meta-learning methods and hard episodes. They found that (i) there is a large gap in performance between easy and hard episodes, (ii) hard episodes are forgotten more easily than easy episodes, (iii) a comparison between adversarial training and curriculum learning strategies show that the former are more effective in improving the performance gap between hard and easy episodes.

**Summary Of The Review:**

In its current form the paper is not ready for acceptance. The main issues are the lack of discussion of out-of-distribution literature and shallow empirical comparisons.

---

> ### Author Response · Authors · 2021-11-20
> **Response to reviewer uYEr**
>
> **“No discussion about out-of-distribution tasks. It seems to me that there is a strong connection between hardness and out-of-distribution samples. Hard episodes could either be out-of-distribution samples or outliers that are on the margin of the decision boundaries. “ and “Limited number of conditions….”** : We thank the reviewer for the suggestion on investigating the connection between episodic hardness and out-of-distribution samples. It is indeed an extremely important and under explored direction in meta-learning. However, we believe that an end to end investigation of the susceptibility of existing meta-learners to OOD tasks warrants a separate work in itself.
>
> We would like to note that the focus of our work is primarily to analyse episode hardness where the meta-training and meta-testing episodes are drawn from the same distribution.
>
> While an end to end investigation of episode hardness and OOD tasks is out of scope for this paper in the limited rebuttal time frame, we have added new analysis and experiments in this regard (See Appendix - Section H). Precisely, we use the perturbations from the ImageNet-C test set to generate new perturbed or OOD episodes during meta-testing.  These new episodes are perturbed with noise, blur etc. In general, we find that such OOD episodes are indeed hard for the meta-learner. For e.g, most of the perturbed episodes have higher losses than the hard in-distribution episodes. Notably, we also find an overlap in the distribution of the query losses incurred by the in-distribution episodes and the OOD episodes. This indicates that some of the hard in-distribution episodes might be outliers or OOD with respect to the task distribution used during meta-training.
>
>
> **“Nature of hard tasks….ORBIT dataset.”**: We thank the reviewer for pointing us towards the ORBIT dataset. We will update the final version of our paper with new experiments on the ORBIT dataset. Precisely, we plan to create meta-testing episodes with different amounts of clutter in the support and query to find how the presence of multiple objects in the frame affect the performance of meta-learners.

---

> > ### Comment · Reviewer_uYEr · 2021-11-24
> > **Answer to rebuttal**
> >
> > Thank you for your answer. I have read the other reviews and checked the new section in Appendix H. The results are interesting and seems to point to a possible overlap between hard and OOD episodes. However, the paper is still lacking in terms of quantitative analysis when it comes down to understanding the nature of hard episodes.
> >
> > In my review I have tried to point to OOD tasks because in practice this is what matters the most in meta-learning. An optimal meta-learner should be able to generalize to tasks under significant shift and this is often the case in OOD episodes. Note that, this has been the focus of recent work, with the community drifting away from CIFAR-FS and miniImageNet towards more realistic benchmarks such as MetaDataset and ORBIT. In other words, I am not convinced that studying hardness in meta-learning is so relevant, especially when it seems so intertwined with OOD.
> >
> > I think that the work is interesting and promising, but the evidences gathered in the paper do not disentangle the factors I have discussed. The main contribution is not substantial enough and the additional results (while encouraging) do not clarify the nature of the problem. Therefore I will not change my score.

---

### Official Review · Reviewer_NN1B · 2021-11-02

**Correctness:** 3
**Technical Novelty And Significance:** 2
**Empirical Novelty And Significance:** 2
**Recommendation:** 3
**Confidence:** 4

**Main Review:**

It is an interesting point that we should not evaluate meta-learners only on average accuracy at test time. I agree with the authors about the need to also understand worse-case behavior before real-world deployment. The findings that there is a large discrepancy between the ‘end’ and ‘max’ accuracy for some episodes is also interesting, and so is the connection to forgetting that is established in this paper.

However, I have a number of concerns about the paper, relating to the motivation of the work, clarity, relationship to related work and experiments. Detailed comments below:

- There is a disconnect between how the work is motivated, and the proposed approach(es). The first paragraph of section 3 criticizes the practice of optimizing the average loss (across episodes) at *training* time (and the proposed approaches address this) but then discusses that this doesn’t give enough insights into the performance when deploying ‘in the wild’ to diverse *test tasks*. There is a disconnect here: If the aim is to gain insights on how meta-learners perform on various *test episodes*, it would be sufficient to simply modify the *evaluation* procedure (e.g. look at the distribution of test performance instead of collapsing to the average). So there seems to be a hidden hypothesis that modifying the *training procedure* will lead to more robust and generally-applicable meta-learners that perform well on diverse *test* tasks? It would be good to state this hypothesis and provide some intuition or evidence for why it’s the case.

- ‘Existing meta-learners … primarily focus on improving prediction performance on average across multiple episodes’. Again, It is not clear whether the criticism relates to the *training objective* (since all training episodes are treated equally without weighted loss usually) or to the *evaluation protocol* (since the evaluation metric is the average over test episodes’ query accuracy).

- ‘We first order the test episodes in decreasing order of hardness’ - how is this done? The provided definition of hardness is specific to a model. Which meta-learner was used for this? Further, ‘we evaluate different meta-learners on the easiest and hardest test episode’: is there a common set of easiest/hardest ones, or does each meta-learner here create its own easy/hard sets? Answering these questions is important for interpreting the results, and unfortunately this information is missing from the paper.

- Fig 1: it would be more informative to show the entire distribution instead of only the min and max test accuracy, as those two points might be extreme outliers.

- Fig 2: which meta-learner was used to obtain the hard and easy episodes visualized here? Are the semantics of hard episodes consistent across meta-learners?

- In Section 5, it’s not clear whether the episodes whose performance is tracked throughout meta-training are training episodes or test episodes. In the first paragraph of Section 5.1: ‘we first randomly select a set of k episodes … and track their accuracy, throughout the course of meta-training’. This sentence does not clarify this. My assumption is that these are *training* episodes in this section (unlike in Section 3), due to the very high accuracy (almost 100%) for easy episodes shown in Figure 3. This should be clarified.

- In section 5.2, the chosen episodes are somehow divided into the hardest and easiest ones. How is this done? Are these the ones that are hardest according to the model *at the last step of training*? If so, it is not too surprising that those don’t perform well at the end of training, by definition of how they were chosen.The large gap between the ‘max’ and ‘end’ for these episodes is still interesting though.

- Which meta-learner was used to produce Figures 3 and 4? Unless I missed it, this isn’t stated anywhere. Further, it would be good to run this analysis for different ones. Do all meta-learners suffer from forgetting hard episodes, or are some more prone to this than others?

- In Fig 5, I was expecting the curves to be monotonically decreasing, since for two thresholds \alpha_1 < \alpha_2, all of the local forgetting events that occurred at threshold \alpha_2 also occur at threshold \alpha_1. Can you help me understand how come this isn’t the case?

- In explaining general adversarial training: ‘Then, for each element of the batch, we additionally sample a number of similar episodes’. What does ‘similar’ mean here? How are similar episodes selected? e.g. same classes but different support/query examples? This needs to be clarified.

- In Table 2, how are hard episodes defined? I assume in this case they’re hard *test* episodes? Is it the same set of episodes that all models are evaluated on? Which meta-learner was used to determine which episodes are hard?

- It would be useful to also experimentally compare with the re-weighted loss method of (Arnold et al. (2021)) that is mentioned in the paper.

- Another closely-related approach is the ‘Hard task (HT) meta-batch’ approach from [1] - a curriculum learning method that schedules hard tasks in meta-training batches by resampling failure cases. They show in that paper that this consistently improves upon random task sampling, in contrast to the curriculum approach explored here.

- Related to the above point, the curriculum learning approach used here is a very simple one, so perhaps it’s premature to claim its failure as a negative result for curriculum learning. For instance, using only easy episodes in the first half and only hard episodes in the second half might not be the right configuration. How were these hyperparameters chosen? Were other design choices explored?

Minor
- In the line below Equation 1, ‘\theta = … is the fine-tuning step …’. I would say that \theta is the parameters resulting from the fine-tuning step, not the fine-tuning step itself. Same comment applies to the sentence below Equation 3.

- ‘we find a strong correlation between the episodic loss and the accuracy (...)’ - some numbers are provided here without explaining what these are. I’m assuming they’re some correlation co-efficients, but what exactly are these?


References
- [1] Meta-Transfer Learning for Few-Shot Learning. Sun et al. CVPR 2019.


**Summary Of The Paper:**

This paper proposes to deviate from studying the average performance of meta-learners on different tasks, and also report separately their performance on ‘hard’ episodes, measuring ‘worst case’ performance which may be critical for deployment in real-world applications. They provide a definition for episode hardness, and investigate the performance on easy and hard episodes. They find that hard episodes are more often forgotten during meta-training compared to easy episodes. They then propose two strategies to mitigate this, based on adversarial and curriculum training. Their experiments on 3 datasets using 3 meta-learners show that the former sometimes leads to improved few-shot learning performance.

**Summary Of The Review:**

Defining and studying hard episodes in meta-learning is an interesting direction that may lead to more robust models that can be more safely deployed in various scenarios in the real world. So this type of study has the potential to be impactful. Certain findings in this paper are also quite interesting, like the large discrepancy between ‘end’ and ‘max’ for certain episodes, and connections to forgetting. However, the paper lacks clarity in certain important areas that makes it difficult to correctly interpret the results. I also found that the paper can be improved in terms of comparisons with related work and more thorough experiments. I believe that another round of review would be required to address these issues, so I do not recommend acceptance at this stage.


#######
Edit: after rebuttal, and reading the other reviews:

I maintain my opinion that the paper, in its current form, falls below the bar for publication. To summarize, albeit some clarity concerns resolved during the rebuttal, I still find the narrative of the paper unclear and the proposed approach not sufficiently well motivated (please refer to my latest response to the authors for details). I also agree with reviewer uYEr that OOD tasks are a natural candidate for ‘hard’ tasks. In fact, an important advantage of defining hardness in this way is that it no longer is dependent on each particular meta-learner. Based on my understanding of the author’s response to my review, the results reported in the tables are computed on different episodes for each meta-learning model, which is problematic for making direct comparisons between them. Finally, I feel that the experimental section is weak and lacks comparisons with relevant previous work. For instance, ‘hard-task meta-batch’ employs a different curriculum learning approach which they find beneficial, whereas the simple curriculum learning proposed here doesn’t work very well, and I felt that they prematurely declare this to be a negative result for curriculum learning. As discussed, running additional experiments to compare against different design choices and related work would strengthen the paper for future revisions.

---

> ### Author Response · Authors · 2021-11-20
> **Response to reviewer NN1B**
>
> We thank the reviewer for appreciating the importance of understanding the worst-case behaviour of meta-learners and also the connections of episode hardness to catastrophic forgetting. We appreciate the reviewers detailed feedback on our work. Below we provide our responses:
>
> **“We first order the test episodes in decreasing order of hardness’ - how is this done? The provided definition of hardness is specific to a model. Which meta-learner was used for this? Further, ‘we evaluate different meta-learners on the easiest and hardest test episode’: is there a common set of easiest/hardest ones, or does each meta-learner here create its own easy/hard sets?”**:  In Section 3.2, the hardest and the easiest episodes were selected specific to the individual meta-learner. For e.g, for R2D2 + ResNet-12, we order the test-episodes based on the query loss incurred with R2D2 specifically to find the easiest and hardest episodes. We also note that (See Appendix (D)), that there is a satisfactory transferability of episode hardness across different meta-learners and architectures.
>
>
> **“Fig 1: it would be more informative to show the entire distribution instead of only the min and max test accuracy, as those two points might be extreme outliers”**: Thank you for the suggestion. In Fig. (9) - Appendix, we have added the plot showing the distribution of episode hardness across different meta-learners for miniImageNet. In the updated draft, we have also added the distribution of episodic accuracy across different meta-learners (R2D2, MetaOptNet) and other datasets such as CIFAR-FS and tieredImageNet (See Appendix (F)).
>
>
> **“In Section 5, it’s not clear whether the episodes whose performance is tracked throughout meta-training are training episodes or test episodes. “**:  We want to clarify that in Section (5), the episodes whose accuracy and hardness measures are tracked throughout the course of meta-training are training episodes.  We have made Section (5) more clear with this detail.
>
> **“In section 5.2, the chosen episodes are somehow divided into the hardest and easiest ones. How is this done? Are these the ones that are hardest according to the model at the last step of training? “**: Amongst the sampled set of episodes, we define the hardness of an episode as the loss incurred on the query examples at the last step of training.  We agree that for hard episodes selected in the way described above, it is expected to observe a low accuracy at the end of meta-training. However we note that, in Section 5.2, the aim is to show that for the set of hard episodes, the underlying meta-learner forgets previously acquired knowledge (denoted by max accuracy in Fig. 3) which is not the case for easy episodes where max accuracy is always reached at the end of meta-training.
>
> **“Which meta-learner was used to produce Figures 3 and 4? Unless I missed it, this isn’t stated anywhere. Further, it would be good to run this analysis for different ones. Do all meta-learners suffer from forgetting hard episodes, or are some more prone to this than others?”**: For Figures 3 and 4, we use prototypical networks with Conv4 backbone to analyze the connections between episode hardness and catastrophic forgetting. We observe a similar behaviour and trend for other meta-learners such as R2D2 and MetaOptNet as well. We ran a similar analysis to compute the forgetting behaviour of episodes on different meta-learners other than prototypical networks such as R2D2 and MetaOptnet and have added the new experimental results in Appendix (G).
>
> **“In explaining general adversarial training: ‘Then, for each element of the batch, we additionally sample a number of similar episodes’. What does ‘similar’ mean here? How are similar episodes selected? “**:   We clarify that by ‘similar’, we mean sampling different support and query examples belonging to the set of classes already sampled in the batch. We have also improved the writing in Section (6) and made the details more clear for the readers.

---

> > ### Author Response · Authors · 2021-11-20
> > **More responses**
> >
> > **“It would be useful to also experimentally compare with the re-weighted loss method of (Arnold et al. (2021)) that is mentioned in the paper” :  and “Another closely-related approach is the ‘Hard task (HT) meta-batch’ approach from [1] - a curriculum learning method that schedules hard tasks in meta-training batches by resampling failure cases. “**:   Both (Arnold et al. (2021) and (Hard task (HT) meta-batch [1]), report only average episodic performances, but do not provide salient insights on how such methods perform on hard episodes. Based on the reviewer’s suggestion, we are running the methods from both ‘Arnold et al. (2021)’ and ‘Hard task meta-batch[1]’ to evaluate their effectiveness on hard episodes. Due to limited time constraints, we will not be able to report comparisons from both these methods across all our experimental settings before the end of the rebuttal period. We will however update the final draft of our paper with these new results. We note that these methods can also serve as strong baselines for performance of meta-learners on hard episodes.
> >
> >
> > **“ There is a disconnect between how the work is motivated, and the proposed approach(es). The first paragraph of section 3 criticizes the practice of optimizing the average loss (across episodes) at training time (and the proposed approaches address this) but then discusses that this doesn’t give enough insights into the performance when deploying ‘in the wild’ to diverse test tasks.”:  and “Existing meta-learners … primarily focus on improving prediction performance on average across multiple episodes’. Again, It is not clear whether the criticism relates to the training objective (since all training episodes are treated equally without weighted loss usually) or to the evaluation protocol”** :   We hypothesize that generalized random sampling and optimizing the average loss during meta-training might be a sub-optimal strategy for good predictive performances on hard episodes. For e.g, we show that a simple adversarial meta-training strategy outperforms the baseline on hard and diverse test episodes. Hence modifying the meta-training strategy is crucial to perform well on hard episodes. In a similar vein, recent and concurrent work such as (Arnold et al. 2021) has shown that a modified episodic sampling procedure via importance sampling also leads to improved average performance, showing evidence that penalizing only the average episodic loss is not the most optimal strategy.
> >
> > With regards to meta-testing and evaluation, current meta-learning methods only report the average episodic performance which does not provide enough insights into how the meta-learner performs on edge cases (i.e, hard episodes) when they are deployed for different applications. In such cases, it’s necessary to look at more robust measures of performance such as the distribution of test-performance.
> >
> > Therefore, our criticism is related to both the training objective as well as the evaluation protocol individually.  Current meta-training objectives fail in generalizing to hard episodes, while the evaluation procedure used during meta-testing fails to generate solid insights about the true performance of the meta-learner under investigation.

---

> > > ### Comment · Reviewer_NN1B · 2021-11-21
> > > **response to authors' rebuttal**
> > >
> > > Thank you for the clarifications. I recommend incorporating in the paper the narrative given in the response to the last point above. Specifically, motivating why looking beyond just the average performance at *test time* is important, and then stating the *hypothesis* that modifying the *training regime* to not simply optimize for the average accuracy over training episodes, is a useful avenue towards achieving this. Further, clarifying for each experiment how the hard episodes were chosen, and whether they’re train or test episodes is really important. I disagree that this is a ‘detail’ as it really impacts the interpretation of the results.
> > >
> > > Based on these clarifications, I have a few more comments.
> > >
> > > - I’m not convinced that doing better at hard episodes at test time requires optimizing for hard episodes at training time, though I agree that the success of adversarial training over vanilla episodic training is a piece of evidence in support of that hypothesis. To explore this more thoroughly and broadly, one missing piece is to also consider non-episodic approaches at training time [1,2] and investigate their performance across the different test episodes. It would be informative to see whether this simpler and effective training paradigm already achieves the goal of performing well on hard test episodes.
> > >
> > > - Now that the authors have clarified that indeed hard episodes are always defined with respect to each meta-learner, I’m concerned about the comparisons in the tables, as this would mean that each model is evaluated on a different set of episodes? It doesn’t seem like it’s possible to make direct comparisons between different models in this setup.
> > >
> > > - I now better understand the proposed adversarial technique based on the clarification of what ‘similar’ episodes meant. One observation is that the classes are still chosen uniformly at random, and the ‘harder’ episodes chosen for training are determined based on the choice of *examples* to populate the support/query sets for each chosen set of classes. This is in contrast to the ‘hard task (HT) meta-batch’ approach mentioned in my original review which is based on just replaying episodes with poor performance, so in that case some *classes* may be seen more often than others during training. I think that the difference in these design choices is interesting and a direct comparison between them would be useful to understand what is most beneficial.
> > >
> > > References
> > > - [1] A Closer Look at Few-shot Classification. Chen et al. ICLR 2019.
> > > - [2] Re-thinking Few-shot Image Classification: A Good Embedding is All you Need? Tian et al. 2020.

---

### Official Review · Reviewer_ceM1 · 2021-11-03

**Correctness:** 4
**Technical Novelty And Significance:** 2
**Empirical Novelty And Significance:** 2
**Recommendation:** 5
**Confidence:** 4

**Details Of Ethics Concerns:**

No concerns.

**Main Review:**

First of all, I would like to appreciate the authors for discussing and comparing against another contemporary work on the same topic (NeurIPS 21) rather than ignoring or hiding the fact that the work is out there.

Strengths
-------------
In this paper, the authors tackle a problem in meta-learning setup that has not received enough recognition -- performance of the meta-learner on easy vs hard tasks in a collection of few-shot tasks (used to train the meta-learner). The authors use a simple quantifiable metric to measure hardness (performance on query samples) and analyze the performance and catastrophic forgetting of various meta-learners on those tasks both during training and at the end. They also discuss two potential improvements which can improve performance on hard episodes - i) adversarial training ii) curriculum learning and show that the first one helps while the second one does not. I found the paper quite enjoyable to read and easy to follow and overall the presentation is nice. The experimental protocol and the ablation studies also complement the narrative well.

Weaknesses
------------------
Although I enjoyed reading the manuscript, I have to say that I do not think the paper at this point, has sufficient novelty either in terms of new technical insights or new experiments. Via some qualitative examples, the authors try to argue on what type if data samples the meta-learner underperform (which eventually lead to hard episodes). However, I would like to see if there is any formal way to identify if an episode will be hard for a meta-learner or not based on the data-points present in support vs query. The qualitative explanations may not hold good for all few-shot tasks or otherwise may require post-hoc deduction as to why it is a hard episode. I would have also liked to see a comparison of hardness for the same task across different learners e.g. if a task is hard for ProtoNet, is it also hard for MetaOptNet? If not, then what is the reason it is not hard for MetaOptNet but for ProtoNet and these type of experiments would be beneficial. Finally, the authors mention that ACT (adversarial + curriculum) performs better than adversarial only on hard episodes - but from table 2, the way I see it, adversarial works better than adversarial + curriculum more often than not.






**Summary Of The Paper:**

In this paper, the authors analyze the hardness of different episodes for an episodic training regime in the context of a meta-learning training (for few-shot classification tasks). The authors show that even though different meta-learners perform relatively similar on average on these tasks, their performances vary significantly when it comes to harder tasks. Authors also showed that the performance on hard tasks can be somewhat improved using an adversarial training strategy.

**Summary Of The Review:**

The paper is nicely written, easy to follow, tackles and under-explored problem and provides a good analysis, but lacks sufficient novelty in methods or experiments and also does not have a strong justification with respect to some of the findings.

---

> ### Author Response · Authors · 2021-11-19
> **Response to reviewer ceM1**
>
> We first thank the reviewer for appreciating that a thorough understanding of episode hardness is an important area in meta-learning.
>
> Below we provide responses to the weaknesses as pointed out by the reviewer.
>
> **I would have also liked to see a comparison of hardness for the same task across different learners e.g. if a task is hard for ProtoNet, is it also hard for MetaOptNet? If not, then what is the reason it is not hard for MetaOptNet but for ProtoNet and these type of experiments would be beneficial** :  Thank you for the suggestion!  We have conducted new experiments to specifically understand the transferability of hard episodes amongst meta-learners. Previously in Appendix (D), we have reported the Pearson and Spearman correlation of episode hardness (i.e, query loss) across different combinations of meta-learners over all the 1000 test-episodes, showing that episode hardness transfers across architectures and meta-learners in general. Across all the episodes we find Pearson correlations ranging from 0.75 to 0.94.
>
> However this analysis does not answer the question : if a subset of episodes are hard for one meta-learner (e.g. Prototypical Networks), are they also hard for another meta-learner (MetaOptNet) also?  To obtain further insights, we first obtain a set of 50 hard episodes from a meta-learner (such as Prototypical Networks) and measure its hardness (in the form of query loss and accuracy) across other meta-learners such as MetaOptNet and R2D2. In summary, we find that although episode hardness transfers in general (See Appendix (D)), such transferability is noisier for hard episodes (See below).
>
> * (Meta-Learner 1, Meta Learner 2 , (Pearson, Spearman))
> * (Proto +ResNet,  R2D2 + ResNet, (0.58,  0.65))
> * (Proto + ResNet , R2D2 + Conv, (0.38, 0.50))
> * (Proto + Conv, R2D2 + Conv, (0.42, 0.428))
> * (Proto + Conv, R2D2 + ResNet,(0.31, 0.24))
> * (Proto +ResNet,  MetaOptNet + ResNet, (0.54,  0.61))
>
>
> **If there is any formal way to identify if an episode will be hard for a meta-learner or not based on the data-points present in support vs query**:   We thank the reviewer for the suggestion. Identifying the hardness of an episode independent of the underlying meta-learner or architecture choice is indeed a challenging and important goal in meta-learning which warrants an independent investigation in itself.  For e.g, knowing the hardness of training episodes apriori can enable us to design more stronger meta-training strategies, taking into account the hardness of episodes as a constraint. To this regard, we make certain initial investigations leveraging self-supervised embeddings and design two baseline experiments:
>
> (a) Experiment 1: Given a training episode, we use a self-supervised learner (SimCLR) trained on ImageNet with ResNet-50 backbone to create class-specific embeddings from the support examples and obtain the query losses via distance to the correct class prototype. Then, we compute the (Pearson, Spearman) correlation of the query losses obtained from SimCLR and the given meta-learner.  We find a correlation of (0.51, 0.52) with the ResNet-12 architecture + and a correlation of (0.52, 0.54) with the Conv-4 architecture for the underlying meta-learner (Prototypical Networks).  We also select a set of 50 hard episodes based on the query losses incurred on the meta-learner (Prototypical Networks + ResNet-12). We then find the correlation of the query losses of these hard episodes with the loss incurred with the self-supervised learner. The (Pearson, Spearman) correlations are (0.49, 0.55).
>
>
> (b) Experiment 2: Given a training episode, we use the self-supervised model (SimCLR) to fine-tune on the support examples with a linear head. Then, we compute the (Pearson, Spearman) correlation of the query losses obtained from SimCLR and the given meta-learner. We obtain a (Pearson, Spearman) correlation of (0.56, 0.53) with the ResNet-12 architecture for the meta-learner (Prototypical Networks). For a set of 50 hard episodes (based on Prototypical Networks + ResNet-12), the (Pearson, Spearman) correlations are (0.53, 0.55).
>
> While the correlations are not extremely strong, we find initial evidence that self-supervised embeddings contain certain information about episode hardness, which can be leveraged to build upon in a future work.  We will add these new results and a discussion section in the Appendix Section in the updated version of the draft.
>
> We however note that identifying the hardness of an episode in an unsupervised way (without knowledge of query labels) is still an open research question and is outside the scope of this current paper.

---

> > ### Author Response · Authors · 2021-11-19
> > **Response to reviewer ceM1**
> >
> > **Finally, the authors mention that ACT (adversarial + curriculum) performs better than adversarial only on hard episodes - but from table 2, the way I see it, adversarial works better than adversarial + curriculum more often than not**:  We would like to clarify that AT (adversarial training) performs better than ACT (adversarial curriculum training)  across most of our experimental settings both on an average as well as on hard episodes (Table 1 and Table 2). Specifically, we note this behaviour in Section 7.2.2 (“ ADVERSARIAL TRAINING IS BETTER THAN CURRICULUM TRAINING” ) where we discuss and present a negative result on the effectiveness of curriculum training on meta-learners.

---

> > > ### Comment · Reviewer_ceM1 · 2021-11-29
> > > **Acknowledge the rebuttal**
> > >
> > > I thank the reviewers for addressing some of my concerns and appreciate the efforts they have put in the short timeframe. However, none of my concerns are clearly addressed at this point. I agree that it may require additional investigation but without it, I do not have enough evidence at this point to increase my score.

---

### Decision · Program_Chairs · 2022-01-20

**Decision:**

Reject

**Comment:**

This paper explores the contrast in performance between easy and hard tasks (episodes) in few-shot image classification and propose mitigating strategies to avoid large performance gaps.

None of the reviewers support the acceptance of this work, despite the authors' detailed rebuttals, with all reviewers confirming their preference for rejection following the author response. Issues raised included lack of clarity of writing and lack of sufficiently convincing experimental results.

I unfortunately could not find a good reason to dissent from the reviewers majority opinion, and therefore also recommend rejection at this time.